# Causal Discovery from Conditionally Stationary Time Series

**Carles Balsells-Rodas** [1]  **Xavier Sumba** [1]  **Tanmayee Narendra** [2]  **Ruibo Tu** [3]
**Gabriele Schweikert** [2]  **Hedvig Kjellström** [3]  **Yingzhen Li** [1]

## Abstract

Causal discovery, i.e., inferring underlying causal relationships from observational data, is highly challenging for AI systems. In a time series modeling context, traditional causal discovery methods mainly consider constrained scenarios with fully observed variables and/or data from stationary time-series. We develop a causal discovery approach to handle a wide class of nonstationary time series that are *conditionally stationary*, where the nonstationary behaviour is modeled as stationarity conditioned on a set of latent state variables. Named State-Dependent Causal Inference (SDCI), our approach is able to recover the underlying causal dependencies, with provable identifiablity for the state-dependent causal structures. Empirical experiments on nonlinear particle interaction data and gene regulatory networks demonstrate SDCI's superior performance over baseline causal discovery methods. Improved results over non-causal RNNs on modeling NBA player movements demonstrate the potential of our method and motivate the use of causality-driven methods for forecasting.

## 1. Introduction

Deep learning has achieved profound success in vision and language modelling (Brown et al., 2020; Nichol et al., 2022). Still, it remains a grand challenge for deep neural networks to perform causal discovery (Yi et al., 2020; Girdhar & Ramanan, 2020; Sauer & Geiger, 2021), which is critical for interpretability, generalization, and robustness (Lake et al., 2017; Schölkopf et al., 2021). Theoretically, structure identifiability is key to ensure a unique correspondence between observations and the underlying causal structures (Peters

et al., 2017). Practically, better algorithms are needed to accurately extract the causal structure from data.

In time series analysis, causal discovery identifies the underlying temporal causal structure of the observed sequences. Historical approaches rely on a restrictive assumption: stationarity (Granger, 1969; Peters et al., 2017; Tank et al., 2021; Löwe et al., 2022), while real-world data is often nonstationary with potential hidden confounders. To address this issue, three major strategies have been recently proposed: (1) modelling nonstationary noise (Huang et al., 2020; Gong et al., 2023); (2) introducing time-dependent effects with a fixed causal structure (Huang et al., 2015; 2019); and (3) using discrete latent variables to capture structural changes over time (Saggioro et al., 2020). Despite these advances, causal discovery in nonstationary time series under realistic assumptions remains an open challenge.

This work addresses causal discovery for nonstationary time series based on a much relaxed assumption, *conditionally stationary time series*, where the dynamics of the observed system change depending on a set of discrete "state" variables. These causal structures can change not only during time, but also across samples (Löwe et al., 2022). This assumption holds for many real-world scenarios, such as individuals whose different decisions depend on mood, past experiences, or interactions with others. The causal discovery task for such conditionally stationary time series poses different challenges depending on the observability and dependency of the states, which we classify into 3 scenarios:

**States observed/independent**: The states are observed and/or independent on observations (Fig. 1a). Structure identifiability can be established for both cases by Peters et al. (2013), and Balsells-Rodas et al. (2024), respectively.

**States determined**: The states are hidden but depend on observations directly. In Fig. 1b, the states are determined by the balls' positions (pink vs purple regions).

**States recurrent**: The states depend on historical events. E.g., in Fig. 1c, particles have state changes upon collision. Also in a football game a player acts differently based on earlier actions of the others.

We propose a novel framework to tackle both the theoretical and algorithmic challenges for causal discovery from condi-

---

[1]Imperial College London [2]University of Dundee [3]KTH Royal Institute of Technology. Correspondence to: Carles Balsells-Rodas <cb221@imperial.ac.uk>.

*Proceedings of the 42$^{nd}$ International Conference on Machine Learning*, Vancouver, Canada. PMLR 267, 2025. Copyright 2025 by the author(s).

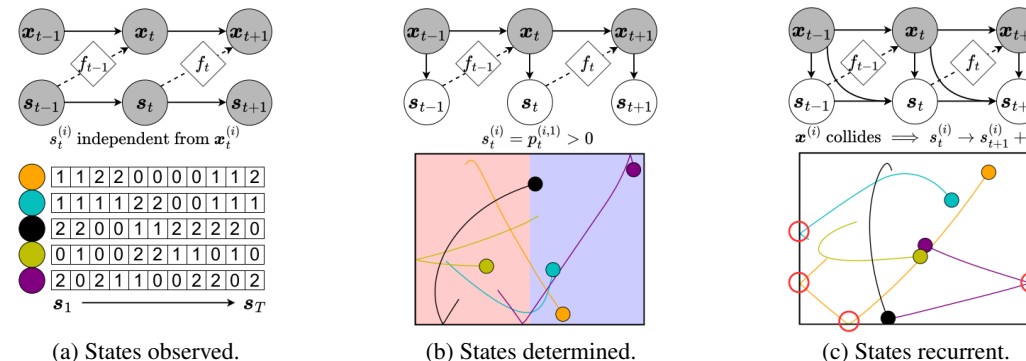

(a) States observed.  (b) States determined.  (c) States recurrent.

Figure 1: Graphical representations of the data generation processes. $\boldsymbol{x}_t$ represents the observations of a time series, and $\boldsymbol{s}_t$ denotes the state variables. The states affect observations by changing the causal structure $f_t$ for different state values.

tionally stationary time series in states determined and recurrent cases. Our contributions are summarised as follows, providing advances in both identifiability and estimation:

- We introduce *conditional summary graph* as a compact representation of the underlying causal graph structure for conditionally stationary time series. This efficiently addresses the exponential complexity of full-time graph in summarising the causal structure of time series data with nonstationary interactions between variables.

- We establish identifiability for the conditional summary graph and related structural properties for conditionally stationary time series satisfying "state determined" (Fig. 1b) or "state recurrent" (Fig. 1c) assumptions.

- We propose State-Dependent Causal Inference (SDCI) as a practical algorithm to extract the conditional summary graphs and model state dependencies, based on discrete latent variable models and graph neural networks.

- We validate SDCI on semi-synthetic data based on physical and biological systems, and real-world datasets. Compared to baselines including GNN and RNN-based approaches, SDCI achieves superior performance in identifying the underlying nonstationary structures in gene regulatory networks; and forecasting future trajectories from observations on NBA player movements.

## 2. Background

### 2.1. Causal Discovery in Stationary Time Series

Causally-driven time series are often modelled using structural causal models (SCM; (Pearl, 2009; Peters et al., 2017)) for describing data generation processes. Consider $N$ sequences of length $T$, denoted by $\boldsymbol{x}_{1:T}$, where $\boldsymbol{x}_t^{(i)} \in \mathbb{R}^d$ denotes the features of the $i$-th sequence at time $t$; which can incorporate high-order moments, e.g velocity, acceleration. The data dependencies in a temporal SCM are structured via a causal graph, known as the *full time graph*, $\mathcal{G}_{1:T}^{FT}$.

For simplicity, we assume no hidden confounders, no in-

stantaneous effects, and a first-order Markov property. In stationary time series, the full time graph is static across time steps. This allows us to define the *summary graph* $\mathcal{G}^S = \{\mathcal{V}, \mathcal{E}^S\}$, where $\mathcal{V} = \{1, \ldots, N\}$ and an edge $i \to j$ exists in $\mathcal{E}^S$ if there is an edge from $\boldsymbol{x}_t^{(i)}$ to $\boldsymbol{x}_{t'}^{(j)}$ in $\mathcal{G}_{1:T}^{FT}$ for some $t < t'$. The identifiability of both $\mathcal{G}_{1:T}^{FT}$ and $\mathcal{G}^S$ is guaranteed under *Time Series Models with Independent Noise* (TiMINo; (Peters et al., 2013)). By further assuming an additive noise model (ANM), the SCM becomes:

$$\boldsymbol{x}_t^{(j)} = f^{(j)} \left( \boldsymbol{x}_{t-1}^{(i)} | \boldsymbol{x}_{t-1}^{(i)} \in \mathbf{PA}^{(j)}(t-1) \right) + \boldsymbol{\varepsilon}_t^{(j)}, \quad (1)$$

where $\mathbf{PA}^{(j)}(t-1)$ denotes the parents[1] of $\boldsymbol{x}_t^{(j)}$ and $\boldsymbol{\epsilon}_t^{(j)} \sim p_{\boldsymbol{\varepsilon}}$ represents independent noise. In this setting, $\mathbf{PA}^{(j)}(t-1)$ is constant in time and aligns with the summary graph $\mathcal{G}^S$.

### 2.2. Markov Switching Models

Markov Switching Models (MSMs; (Hamilton, 1989)) extend time series modeling by introducing discrete latent variables $u_t \in \{1, \ldots, U\}$ that condition the autoregressive process at each time step $t$. For *regime-dependent time series* (Saggioro et al., 2020), the full-time graph $\mathcal{G}_{1:T}^{FT}$ is time-dependent, but *causally stationary* (Assaad et al., 2022) given the discrete latent variables $u_t, t \in \{1, \ldots, T\}$. This leads to defining the *regime-dependent graph*, which is a set of graphs $\mathcal{G}_{1:U}^{RD} := \{\mathcal{G}_u^{RD} | 1 \leq u \leq U\}$ where $\mathcal{G}_u^{RD}$ encodes the causal effects of $\boldsymbol{x}_t$ for $u_t = u$. Our work utilises MSMs under the following key assumptions (m1-m3):

**(m1) Conditional first-order Markov transitions**, controlled by $u_{t-1}$ only: for any $t \in \{2, ..., T\}$,

$$p(\boldsymbol{x}_t | \boldsymbol{x}_{1:t-1}, \boldsymbol{u}_{1:t-1}) = p(\boldsymbol{x}_t | \boldsymbol{x}_{t-1}, u_{t-1}). \quad (2)$$

**(m2) Conditional stationarity:** the transition distributions do not change during time: for any $u \in \{1, ..., U\}$ we have

---

[1]The notation $(t-1)$ indicates that the parents of variable $j$ are considered at the previous time step.

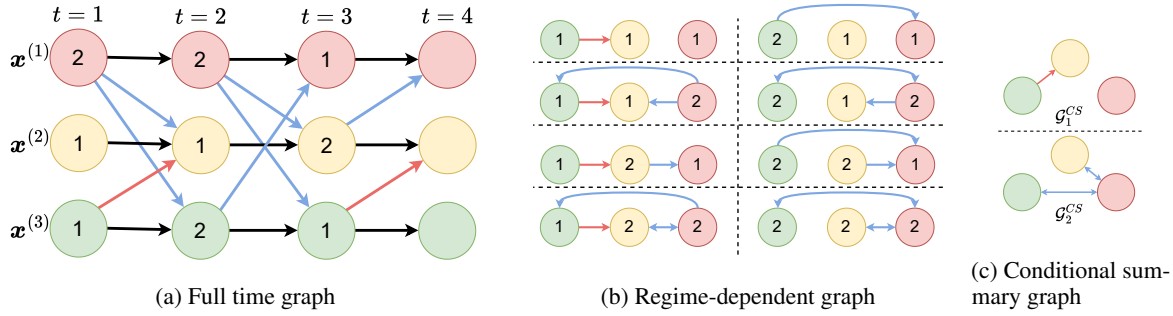

(a) Full time graph

(b) Regime-dependent graph

(c) Conditional summary graph

Figure 2: (a) Full time graph $\mathcal{G}_{1:T}^{FT}$ of a sample considering our problem setting, (b) regime-dependent graph, and (c) conditional summary graph $\mathcal{G}_{1:K}^{CS}$ of the corresponding sample for $K = 2$. We denote states using numbers $\{1, 2\}$ inside each element. Different colors (red and blue) denote effects caused by different states.

for any $t \neq t'$, any $\boldsymbol{\beta}, \boldsymbol{\gamma} \in \mathbb{R}^{Nd}$ such that

$$p(\boldsymbol{x}_t = \boldsymbol{\beta}|\boldsymbol{x}_{t-1} = \boldsymbol{\gamma}, u_{t-1} = u) =$$
$$p(\boldsymbol{x}_{t'} = \boldsymbol{\beta}|\boldsymbol{x}_{t'-1} = \boldsymbol{\gamma}, u_{t'-1} = u). \quad (3)$$

**(m3) Factorisation based on a Gaussian ANM:**

$$p(\boldsymbol{x}_t|\boldsymbol{x}_{t-1}, u_{t-1}) = \prod_{j=1}^{N} \mathcal{N}\Big(\boldsymbol{x}_t^{(j)};$$

$$f_{u_{t-1}}^{(j)}\big(\boldsymbol{x}_{t-1}^{(i)}|\boldsymbol{x}_{t-1}^{(i)} \in \mathbf{PA}_{u_{t-1}}^{(j)}(t-1)\big), \sigma^2 \mathbf{I}\Big), \quad (4)$$

for any $t \in \{2, ..., T\}$, where $\mathbf{PA}_{u_{t-1}}^{(j)}(t-1)$ denotes the parents of variable $j$ at time $t-1$, as described by $\mathcal{G}_{u_{t-1}}^{RD}$.

## 3. State-Dependent Causal Inference (SDCI)

### 3.1. Conditionally Stationary Time Series

We focus on nonstationary time series where each time step $t$ is associated with $N$ latent variables $\boldsymbol{s}_t = \{s_t^{(1)}, \ldots, s_t^{(N)}\}$. Each $s_t^{(i)} \in \{1, ..., K\}$ controls the causal effects of $\boldsymbol{x}_t^{(i)}$ to future observations $\boldsymbol{x}_{t+1}$. In other words, the causal influences of $\boldsymbol{x}_t^{(i)}$ are dynamically modified by the state $s_t^{(i)}$. This can be viewed as a general MSM under assumptions (m1-m3) by setting $u_t = \varphi(\boldsymbol{s}_t)$ with global states $U = K^N$, for some injective $\varphi : \{1, \ldots, K\}^N \to \{1, \ldots, U\}$. Fig. 2a exemplifies the full time graph illustrating these assumptions. Here, the causal effects vary across time as $\mathbf{s}_1 \neq \mathbf{s}_2 \neq \mathbf{s}_3$, leading to nonstationarity. From a MSM view, we can extract a regime-dependent graph with $K^N$ regimes (Fig 2b).

Although the regime-dependent graph captures the general structure, it becomes inefficient as it scales exponentially with $N$. Furthermore, a summary graph (aggregating all the regimes) can be non-informative, due to inability to distinguish state-dependent effects. Instead, we define the *conditional summary graph*.

**Definition 3.1** (Conditional summary graph, first-order Markov setting). The *conditional summary graph* is a set

of $K$ graphs, $\mathcal{G}_{1:K}^{CS} = \{\mathcal{G}_k^{CS}|1 \leq k \leq K\}$, where $K$ is the number of possible state values. Each summary graph $\mathcal{G}_k^{CS} := \{\mathcal{V}, \mathcal{E}_k^{CS}\}$ has the same vertices $\mathcal{V} = \{1, \ldots, N\}$. An edge $i \to j$ exists in $\mathcal{E}_k^{CS}$ if there exists $t$ such that $s_t^{(i)} = k$ and $\boldsymbol{x}_t^{(i)}$ connects to $\boldsymbol{x}_{t+1}^{(j)}$ in $\mathcal{G}_{1:T}^{FT}$.

For simplicity, we omit self-connections in Fig. 2c (black edges in Fig. 2a), where we show the conditional summary graph extracted from the full time graph. For $k = 1$ we observe $s_1^{(3)} = 1$, and a "red edge" connects $\boldsymbol{x}_1^{(3)}$ and $\boldsymbol{x}_2^{(2)}$, placing a red edge in $\mathcal{E}_1^{CS}$. Similar reasoning applies for $\mathcal{G}_2^{CS}$. Compared to summary or regime-dependent graphs (Fig. 2b), the conditional summary graph offers a compact and informative representation of state-dependent causal structures. In summary, the SEM for conditionally stationary time series can be formalised as follows:

$$\boldsymbol{x}_t^{(j)} = f_{\boldsymbol{s}_{t-1}}^{(j)}\left(\boldsymbol{x}_{t-1}^{(i)}|\boldsymbol{x}_{t-1}^{(i)} \in \mathbf{PA}_{\boldsymbol{s}_{t-1}}^{(j)}(t-1)\right) + \varepsilon_t^{(j)}. \quad (5)$$

This introduces the following assumption:

**(m4) Multi-state dependence:** The parents $\mathbf{PA}_{\boldsymbol{s}_{t-1}}^{(j)}(t-1)$ of variable $j$ at time $t$ are defined by $\mathcal{G}_{1:K}^{CS}$ as:

$$\mathbf{PA}_{\boldsymbol{s}_{t-1}}^{(j)}(t-1) := \Big\{\boldsymbol{x}_{t-1}^{(i)}|$$
$$j \in \mathcal{C}_k^{(i)}, k = s_{t-1}^{(i)}, 1 \leq i \leq N\Big\}. \quad (6)$$

$\mathcal{C}_k^{(i)} \subseteq \mathcal{V}$ denotes the children of variable $i$ in state $k$, specified by $\mathcal{G}_k^{CS}$. To illustrate, in Fig. 2a the parents of $\boldsymbol{x}_2^{(2)}$ are $\{\boldsymbol{x}_1^{(1)}, \boldsymbol{x}_1^{(2)}, \boldsymbol{x}_1^{(3)}\}$ as $2 \in C_{s_1^{(1)}}^{(1)}$ and $2 \in C_{s_1^{(3)}}^{(3)}$.

### 3.2. Amortisation of Conditional Effects.

The conditional summary graph introduces linear scaling with $K$. However, indexing the SEM in Eq. (5) by $\boldsymbol{s}_{t-1}$ requires $K^N$ functions. Taking inspiration from interactive systems (Kipf et al., 2018; Löwe et al., 2022), we propose a two-stage interaction and aggregation framework, formalised as the following assumption:

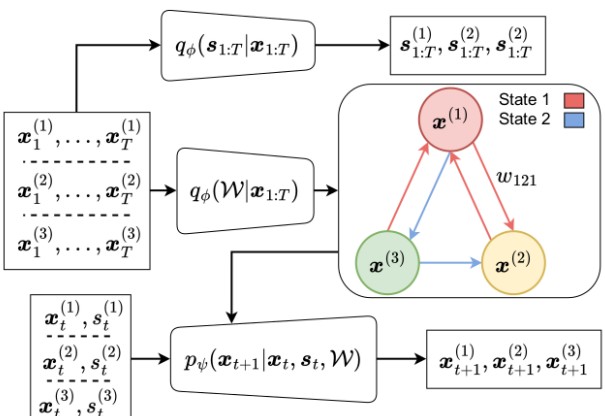

Figure 3: SDCI extracts the *labels of conditional effects* that describe state-dependent interactions in a sample.

**(m5) Global function-type dependence:** The effect between any pair of variables is determined by $n_\epsilon$ functional effects $\mathcal{F} := \{f_0, \ldots, f_{n_\epsilon-1}\}$, where $f_0(\cdot) = \mathbf{0}$ represents the absence of an edge. Each functional effect is state-dependent, i.e., the effects from variable $i$ at time $t$ are determined by its state $s_t^{(i)}$. These effects are collected in the *labels of conditional effects*:

$$\mathcal{W} := \Big\{ w_{i,j,k} \in \{0, \ldots, n_\epsilon - 1\} | 1 \le i, j \le N,$$
$$1 \le k \le K, w_{i,j,k} = 0 \iff j \notin \mathcal{C}_k^{(i)} \Big\}, \quad (7)$$

where $w_{i,j,k}$ specifies the edge-type $i \to j$ at state $k$, associated to variable $i$. In the aggregation stage, the interactions are combined using a permutation invariant function $g$:

$$f_{\boldsymbol{s}_{t-1}}^{(j)}(\boldsymbol{x}_{t-1}) := g\Big( \boldsymbol{x}_{t-1}^{(j)},$$
$$\Big\{ f_{e_i}\Big( \boldsymbol{x}_{t-1}^{(i)}, \boldsymbol{x}_{t-1}^{(j)} \Big) | i \ne j \Big\} \Big), \quad (8)$$

where $e_i = w_{i,j,s_{t-1}^{(i)}}$. The function $g$ aggregates interactions to variable $j$, capturing complex state-dependent dynamics. This formulation effectively reduces the exponential complexity of conditionally stationary time series, and can be efficiently implemented with graph neural networks (GNNs) (Kipf et al., 2018). Furthermore, it aligns with well-established mechanisms in physical systems, such as aggregating forces from pair-wise interactions.

### 3.3. Implementation

We propose State-Dependent Causal Inference (SDCI), a probabilistic approach to infer the underlying causal structure from time series. Given a dataset $\mathcal{D}$, each sample $\boldsymbol{x}_{1:T} \sim \mathcal{D}$ is driven by $\mathcal{W}$, which may differ across samples. SDCI models the joint dynamics of states, edge-types, and observations (Fig. 3), and builds on graph neural networks (GNNs) and interactive systems (Kipf et al., 2018).

**Generative model.** The joint distribution for conditionally stationary time series is dependent on $\boldsymbol{s}_{1:T}$, $\mathcal{W}$, and $\psi :=$

$\{\psi_w, \psi_x, \psi_s\}$. For observed states, we define:

$$p_\psi(\boldsymbol{x}_{1:T}, \mathcal{W} | \boldsymbol{s}_{1:T}) = p_{\psi_x}(\boldsymbol{x}_{1:T} | \boldsymbol{s}_{1:T}, \mathcal{W}) p_{\psi_w}(\mathcal{W}), \quad (9)$$

with a factorised prior on the edge labels $p_{\psi_w}(\mathcal{W}) = \prod_{k=1}^K \prod_{ij} p_{\psi_w}(w_{ijk})$. We can guide $\psi_w$ through domain knowledge (e.g. sparsity). Given $\mathcal{W} \sim p_{\psi_w}(\mathcal{W})$ and $\boldsymbol{s}_{1:T}$, a sequence $\boldsymbol{x}_{1:T}$ is generated as

$$p_{\psi_x}(\boldsymbol{x}_{1:T} | \boldsymbol{s}_{1:T}, \mathcal{W}) = \prod_{t=0}^{T-1} \prod_{j=1}^N p_{\psi_x}(\boldsymbol{x}_{t+1}^{(j)} | \boldsymbol{x}_t, s_t, \mathcal{W}).$$

To compute $p_{\psi_x}(\boldsymbol{x}_{t+1}^{(j)} | \boldsymbol{x}_t, \boldsymbol{s}_t, \mathcal{W})$, the model queries edge-types $e_t^{(ij)} = w_{ijk'}$ for $s_t^{(i)} = k'$, retrieves pairwise interactions $f_e(\boldsymbol{x}_t^{(i)}, \boldsymbol{x}_t^{(j)})$, and aggregates them:

$$\mathbf{h}_t^{(ij)} = \sum_{e>0} \mathbf{1}_{\big( e_t^{(ij)} = e \big)} f_e(\boldsymbol{x}_t^{(i)}, \boldsymbol{x}_t^{(j)}),$$
$$\tilde{\boldsymbol{x}}_{t+1}^{(j)} = \boldsymbol{x}_t^{(j)} + g\Big( \sum_{i \ne j} \mathbf{h}_t^{(ij)}, \boldsymbol{x}_t^{(j)} \Big), \quad (10)$$

where $\mathcal{F} := \{f_e\}_{e=1}^{n_\epsilon - 1}$ and $g$ are parametrisable functions, similar to Eq. (8). $\tilde{\boldsymbol{x}}_{t+1}^{(j)}$ denotes the mean of $\boldsymbol{x}_{t+1}^{(j)}$, which is Gaussian distributed with covariance $\sigma^2 I$. For latent states with influence from $\boldsymbol{x}_{1:T}$ (determined and recurrent cases; Figs. 1b, 1c), the joint distribution extends to:

$$p_\psi(\boldsymbol{x}_{1:T}, \boldsymbol{s}_{1:T}, \mathcal{W}) = p_{\psi_w}(\mathcal{W})$$
$$\prod_{t=1}^T p_{\psi_x}(\boldsymbol{x}_t | \boldsymbol{x}_{t-1}, \boldsymbol{s}_{t-1}, \mathcal{W}) p_{\psi_s}(\boldsymbol{s}_t | \boldsymbol{x}_{t:t-L_x}, \boldsymbol{s}_{t-1:t-L_s}).$$

where $L_x$ and $L_s$ are the maximum lags. The determined case fixes $L_x = 0$ and $L_s = 0$; and we assume the states are autonomous with shared parameters $\psi_s$:

$$p_{\psi_s}(\boldsymbol{s}_t | \boldsymbol{x}_{t:t-L_x}, s_{t-1:t-L_s}) =$$
$$\prod_{i=1}^N p_{\psi_s}(\boldsymbol{s}_t^{(i)} | \boldsymbol{x}_{t:t-L_x}^{(i)}, \boldsymbol{s}_{t-1:t-L_s}^{(i)}). \quad (11)$$

**Inference.** Building on VAE-based approaches (Kingma & Welling, 2014; Löwe et al., 2022), we introduce a variational posterior parametrised by $\phi := \{\phi_w, \phi_s\}$; which approximates the posterior over $\mathcal{W}$, and $\boldsymbol{s}_{1:T}$ as follows:

$$q_\phi(\mathcal{W}, \boldsymbol{s}_{1:T} | \boldsymbol{x}_{1:T}) = q_{\phi_w}(\mathcal{W} | \boldsymbol{x}_{1:T}) q_{\phi_s}(\boldsymbol{s}_{1:T} | \boldsymbol{x}_{1:T}). \quad (12)$$

where $q_{\phi_w}$ is factorised across edges

$$q_{\phi_w}(w_{ijk} | \boldsymbol{x}_{1:T}) = \text{softmax}(\phi_{ijk} / \tau),$$
$$\phi_{ij} = f_{\phi_w}(\boldsymbol{x}_{1:T})_{ij} \in \mathbb{R}^{K \times n_\epsilon}, \quad (13)$$

The function $\phi_{ij}$ extracts embeddings for any pair $i \to j$ that represents the state-dependent causal interaction, and the architecture of $f_{\phi_w}(\boldsymbol{x}_{1:T})$ is based on (Chen et al., 2021). See Appendix B.3 for details. In the determined case, exact

inference on the states is tractable (see Appendix B.2), and we set $q_{\phi_s}(\boldsymbol{s}_{1:T}|\boldsymbol{x}_{1:T}) = p_{\psi_s}(\boldsymbol{s}_{1:T}|\boldsymbol{x}_{1:T})$.

$$q_{\phi_s}(s_t^{(i)}|\boldsymbol{x}_t^{(i)}) = \text{softmax}(\hat{s}_t^{(i)}/\gamma), \quad \hat{s}_t^{(i)} = f_{\phi_s}(\boldsymbol{x}_t^{(i)}), \quad (14)$$

In the recurrent case, $q_{\phi_s}(\boldsymbol{s}_{1:T}|\boldsymbol{x}_{1:T}) = \prod_{t=1}^{T} q_{\phi_s}(\boldsymbol{s}_t|\boldsymbol{x}_{1:T})$ is implemented via a bidirectional RNN-GNN that approximates smoothing (details in Appendix B.3).

**Objective.** SDCI optimises the ELBO (Kingma & Welling, 2014) (see Appendix B.1 for derivation):

$$\log p_\psi(\boldsymbol{x}_{1:T}) \geq -KL\Big(q_{\phi_w}(\mathcal{W}|\boldsymbol{x}_{1:T})\Big\|p_{\psi_w}(\mathcal{W})\Big)$$

$$-\sum_{t=1}^{T} KL\Big(q_{\phi_s}(\boldsymbol{s}_t|\boldsymbol{x}_{1:T})\Big\|p_{\psi_s}(\boldsymbol{s}_t|\boldsymbol{x}_{t:t-L_x}, \boldsymbol{s}_{t-1:t-L_s})\Big)$$

$$+ \mathbb{E}_{q_\phi(\mathcal{W},\boldsymbol{s}_{1:T}|\boldsymbol{x}_{1:T})}\Big[\log p_{\psi_x}(\boldsymbol{x}_{1:T}|\boldsymbol{s}_{1:T},\mathcal{W})\Big], \quad (15)$$

where reparametrisation tricks (Jang et al., 2017) ensure backpropagation for discrete samples $\mathcal{W}, \boldsymbol{s}_{1:T} \sim q_\phi(\mathcal{W}, \boldsymbol{s}_{1:T}|\boldsymbol{x}_{1:T})$. To reduce variance, we use straight-through Gumbel-softmax and fix gradients to pass through unperturbed marginals (Ahmed et al., 2023). During training, Eq. (10) becomes:

$$\mathbf{h}_t^{(ij)} = \sum_{k=1}^{K} \mathbf{1}_{(s_t^{(i)}=k)} \sum_{e>0} \mathbf{1}_{(w_{ijk}=e)} f_e(\boldsymbol{x}_t^{(i)}, \boldsymbol{x}_t^{(j)}), \quad (16)$$

and we optimise $\phi$ and $\psi$ using mini-batch gradient ascent.

# 4. Theoretical Considerations

We provide identifiability guarantees for the labels of conditional effects $\mathcal{W}$, driving conditionally stationary time series under assumptions (m1-m5). A detailed analysis of identifiability for regime-dependent graphs under (m1-m3) and conditional summary graphs under (m1-m4) is provided in Appendix A. We first define partial identifiability for $\mathcal{W}$.

**Definition 4.1** (Partial identifiability of conditional effects). Given observations $\boldsymbol{x}_{1:T}$ and a family of models $\mathcal{M}$ satisfying (m1-m5), we say $\mathcal{M}$ is *partially identifiable with respect to the labels of conditional effects* if for any $p, \hat{p} \in \mathcal{M}$, with corresponding $\mathcal{W}$ and $\widehat{\mathcal{W}}$ such that $p(\boldsymbol{x}_{1:T}) = \hat{p}(\boldsymbol{x}_{1:T})$; $K = \hat{K}$ and there exist permutations $\pi \in S_{n_\epsilon}$ and $\sigma^{(i)} \in S_K$ such that for any $i, j \in \{1, \ldots, N\}$ and $k \in \{1, \ldots, K\}$:

$$w_{i,j,k} = \hat{w}_{i,j,\sigma^{(i)}(k)} \iff w_{i,j,k} = 0, \quad (17)$$

$$w_{i,j,k} = \pi\left(\hat{w}_{i,j,\sigma^{(i)}(k)}\right) \iff w_{i,j,k} \neq 0. \quad (18)$$

*Remark* 4.2. Mixture models can only be identified up to permutations (Yakowitz & Spragins, 1968). Contrary to previous work (Gassiat et al., 2016; Hälvä & Hyvarinen, 2020; Song et al., 2024), our results do not require knowledge of the number of components $K$. Eq. (17) establishes equivalences of $\mathcal{G}_{1:K}^{SG}$ up to element-wise permutations $\sigma^{(i)}$ for outgoing edges $\{C_1^{(i)}, \ldots, C_K^{(i)}\}$. Eq. (18) ensures permutation equivalence in edge labels $1, \ldots, n_\epsilon$.

This partial identifiability definition excludes the state distribution, as no restrictions on the state dynamics are imposed. It cannot generally be achieved under (m1-m5) without further assumptions. However, our main focus in this work is to recover state-dependent structures, and we leave state distribution identifiability for future work. Below, we list sufficient conditions for identifiability.

**(a1) Unique indexing of outgoing structure.** For each state $s_{t-1}^{(i)} \in \{1, ..., K\}^N$, the graph representing the direct causes of $i$ is unique. That is, for any $i \in \{1, \ldots, N\}$:

$$\forall k, k' \in \{1, ..., K\}, k \neq k' \quad \Leftrightarrow \quad \mathcal{C}_k^{(i)} \neq \mathcal{C}_{k'}^{(i)}. \quad (19)$$

This is stronger than the typical unique indexing assumption for mixture models (Balsells-Rodas et al., 2024), where:

$$p(\boldsymbol{x}_t|\boldsymbol{x}_{t-1}, \boldsymbol{s}_{t-1} = (k_1, \ldots, k_N)) \neq$$
$$p(\boldsymbol{x}_t|\boldsymbol{x}_{t-1}, \mathbf{s}_{t-1} = (k_1', \ldots, k_N')),$$

for $(k_1, \ldots, k_N) \neq (k_1', \ldots, k_N')$.

**(a2) Unique indexing of function types.** Given (m1-m5):

**(a2.1):** For any $j \in \{1, \ldots, N\}$, and given $g(\boldsymbol{x}^{(j)}, \{\boldsymbol{h}^{(i,j)}|i \neq j\})$, where $\boldsymbol{h}^{(i,j)} = f_e(\boldsymbol{x}^{(i)}, \boldsymbol{x}^{(j)})$ for some $e \in \{0, \ldots, n_\epsilon - 1\}$, the partial derivative $\frac{\partial g}{\partial \boldsymbol{h}^{(i,j)}}$ is non-zero almost everywhere for any $i \in \{1, \ldots, N\}$.

**(a2.2):** Edge-types differ almost everywhere: $f_0 := \mathbf{0}$, and for $e \neq e', \in \{0, n_\epsilon - 1\}$, the following set has zero measure:

$$\mathcal{X}_{e,e'} := \Bigg\{ \boldsymbol{x}_1 \in \mathbb{R}^d : \exists \boldsymbol{x}_2 \in \mathbb{R}^d, $$
$$\frac{\partial f_e(\boldsymbol{x}_1, \boldsymbol{x}_2)}{\partial \boldsymbol{x}_1} = \frac{\partial f_{e'}'(\boldsymbol{x}_1, \boldsymbol{x}_2)}{\partial \boldsymbol{x}_1} \Bigg\}. \quad (20)$$

These assumptions ensure pairwise interactions are not cancelled during aggregation, and the edge-types remain sufficiently distinct. For example, SDCI, naturally satisfies (a2.1) through GNN message passing. Additionally, implementing edge-type functions as analytic functions guarantees (a2.2). We now state the following identifiability result:

**Theorem 4.3.** *Conditionally stationary time series satisfying (m1-m5) are partially identifiable with respect to conditional effects (Def. 4.1), if they meet assumptions of unique indexing of (a1) outgoing structure and (a2) function types.*

*Proof sketch.* The full proof is depicted in Appendix A.5, and it can be divided into 3 major steps.

(i) Write $p(\boldsymbol{x}_t|\boldsymbol{x}_{1:t-1})$ as a mixture model. Under (m1-m3) and (a1-a2), the regime-dependent graph is identifiable up to a permutation $\tau \in S_{K^N}$ (Theorem A.6). Under state dependencies on $\boldsymbol{x}_{1:t-1}$, the key is to show that the permutation equivalence of the transition distribution is preserved almost everywhere due to (a1-a2), no matter the overlap on $\boldsymbol{x}_{t-1}$.

(ii) Using (m4) and (a1), regime-dependent graph identifiability transfers to conditional summary graph identifiability (Theorem A.7).

(iii) By (m5) and (a1-a2), partial derivatives in the aggregation step reveal permutation equivalence on the edge-types (Corollary A.8).

While several prior works (Hälvä & Hyvarinen, 2020; Song et al., 2024; Balsells-Rodas et al., 2024) introduce nonstationarity via regime switching, our approach offers three key advantages:

- **Unknown number of state values** $K$: Most prior work, including HMM-ICA (Hälvä & Hyvarinen, 2020) and CtrlNS (Song et al., 2024), assumes that the number of latent regimes is known. Our approach leverages Yakowitz & Spragins (1968), which enables identifiability without knowing the true number of regimes. This allows identification of $K$ by model selection (McLachlan & Peel, 2000) (assuming convergence to the MLE), which is not theoretically supported when $K$ requires to be known.

- **No assumption on state dependency:** Prior works typically assume either the "states independent" case (Hälvä & Hyvarinen, 2020; Balsells-Rodas et al., 2024; Rahmani & Frossard, 2025); or the "states determined" case under strong structural constraints (Song et al., 2024). Our identifiability results make no assumptions about state dependencies, allowing feedback from observations.

- **Regime-dependent identifiability:** Our theoretical results (Thms. A.6-A.7; Cor. A.8) build from general Markov Switching Models to our specific implementation, SDCI. Theorem A.6 introduces a novel proof strategy that extends from Yakowitz & Spragins (1968), providing a general theoretical foundation for regime-dependent causal discovery (Saggioro et al., 2020).

As mentioned, SDCI considers samples $\boldsymbol{x}_{1:T} \sim \mathcal{D}$, each generated under different structures. The presented results remain valid, as identifiability holds even when considering the distribution of a single sample. However, the consistency of SDCI, along with other approaches within the same family, such as ACD (Löwe et al., 2022), remains an open challenge. Recent works (Gong et al., 2023; Geffner et al., 2024) have shown that the validity of variational objectives for causal discovery relies on strong assumptions; including universal approximation properties of $q$, absence of model misspecification, and the infinite data limit. These works assume fixed structures across samples, where SDCI can be adapted by setting $q_{\phi_w}(\mathcal{W}|\boldsymbol{x}_{1:T}) = q_{\phi_w}(\mathcal{W})$.

## 5. Related Work

Causal discovery for time series traditionally extends from non-temporal data (Assaad et al., 2022). These approaches include constraint-based methods (Entner & Hoyer, 2010; Runge, 2018), score-based methods such as DYNOTEARS (Pamfil et al., 2020) based on Dynamic Bayesian Networks (Murphy et al., 2002), and functional causal model-based methods (Shimizu et al., 2006; Zhang & Hyvärinen, 2009; Peters et al., 2014) with constrained dynamics assumptions (Peters et al., 2013). The functional causal model-based approaches also motivate the use of deep generative models for causal discovery in high-dimensions (Yu et al., 2019).

Our work focuses on modelling nonstationary time series with discrete latent variables that condition the underlying structure. It is related to Amortized Causal Discovery (ACD; Löwe et al. (2022)), which assumes stationarity and amortizes summary graph discovery across samples with different graphs but shared dynamics. We extend ACD by allowing the causal structure to vary according to latent state variables, thus enabling conditional stationarity. When observed auxiliary variables are present, Monti et al. (2020) uses time contrastive learning (Hyvarinen & Morioka, 2016) and nonlinear ICA for causal discovery; while Yao et al. (2022) establishes identifiability conditions in the presence of piecewise stationary disturbances. Regime-dependent dynamics assume no access to states Saggioro et al. (2020), where Balsells-Rodas et al. (2024) proves identifiability through Markov Switching Models; and Rahmani & Frossard (2025) via a score-based approach. Varambally et al. (2024) also proves identifiability when the discrete latents represent global states. Our "states determined" model is similar to Song et al. (2024), where identifiability is established by assuming mechanism sparsity and knowledge of the number of regimes. Other works focus on fixed causal structures, modelling nonstationarity through noise distributions (Huang et al., 2020; Gong et al., 2023) or time-varying effects (Huang et al., 2015; Zhang et al., 2017; Ghassami et al., 2018; Huang et al., 2019); such as Huang et al. (2020) with distribution shifts; Gong et al. (2023) with history-dependent noise; or Huang et al. (2015) with time-varying functional causal models.

## 6. Experiments

### 6.1. Springs and Magnets

We evaluate SCDI on *spring data* adapted from Kipf et al. (2018); Löwe et al. (2022), consisting of particles in a box, connected by springs with directed impact – meaning that e.g. particle $i$ could affect particle $j$ with a force through a connecting spring, but leaving particle $i$ unaffected by this spring force. We further modify the dataset to introduce repulsive magnetic interactions, resulting in 3 edge-types: none, spring, and magnetic. We assess SDCI across determined and recurrent states. In the determined case, the state is modeled as a function of particles' positions, while the recurrent case involves state transitions when particles

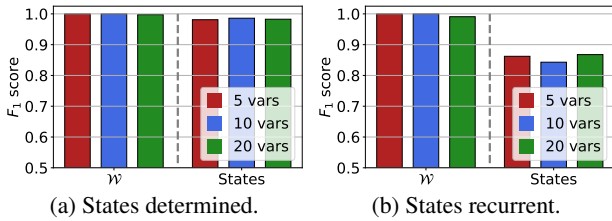

Figure 4: $F_1$ Score of the edge labels $\mathcal{W}$ and states for samples with a shared underlying structure with $K = 2$.

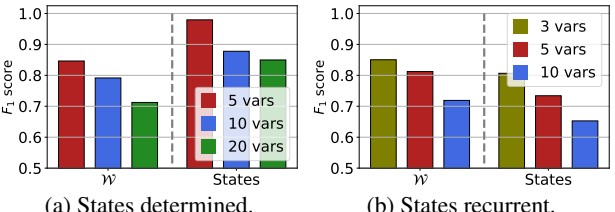

Figure 5: $F_1$ Score of the edge labels $\mathcal{W}$ and states with structures varying across samples with $K = 2$.

collide with walls. Details on data generation, model hyperparameters, and additional results (including observed case) are provided in Appendices C.1, B.4, and E.1 respectively.

To empirically verify the consistency of SDCI in estimating state-dependent structures, we begin with a fixed-graph setting, where the graph remains constant across samples, and use an amortised posterior $q_{\phi_w}(\mathcal{W}|\boldsymbol{x}_{1:T}) = q_{\phi_w}(\mathcal{W})$ for inference. Fig. 4 demonstrates that SDCI achieves perfect structure estimation on both state settings. However, accurately estimating the states in the recurrent case is more challenging, likely due to the exponential computational cost of exact inference. Our formulation considers graphs that vary across samples, and we present results again in both state cases in Fig. 5. In this case, recovering state-dependent structures is significantly more challenging. While SDCI successfully captures the state dynamics in the determined case, the increased complexity from variable graph structures makes recovering state-dependent structures in the recurrent case notably difficult, even with only 3 variables.

We compare SDCI with causal discovery baselines in extracting graphs from conditionally stationary time series. Specifically, we measure the $F_1$ score for summary graph identification in determined states data with fixed and variable graph structures, benchmarking against ACD (Löwe et al., 2022), Rhino (Gong et al., 2023), and Neural Granger Causality (N-GC; (Tank et al., 2021)). For variable graph datasets, Rhino and N-GC require re-training for each sample, adding computational overhead. Results in Fig. 6 show a clear advantage of SDCI in terms of summary graph estimation. We observe that approaches assuming causal stationarity struggle to aggregate the full structure into a coherent summary graph, limiting their performance. How-

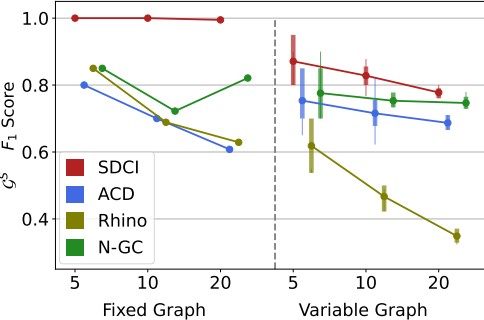

Figure 6: Summary graph ($\mathcal{G}^S$) $F_1$ score on determined states data, with fixed and variable structures across samples.

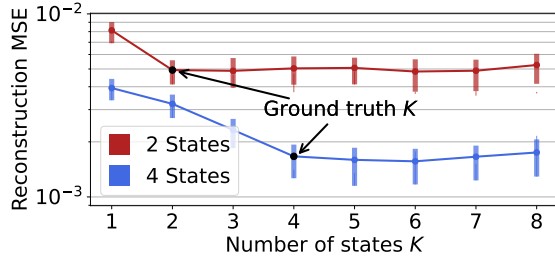

Figure 7: Test reconstruction MSE vs number of states $K$ for data with ground truth $K = 2$ (red) and $K = 4$ (blue).

ever, N-GC particularly achieves strong results for variable graphs, surpassing ACD despite its simplicity. Overall, SDCI decomposes the nonstationary dynamics into conditional stationary components while accurately capturing state transition dynamics.

**Selecting the number of states.** Determining the number of states $K$, or the number of regimes in regime-switching dynamics, is a fundamental challenge in real-world data. As discussed in Section 4, SDCI is also identifiable in terms of $K$. Assuming the consistency of SDCI, $K$ can be theoretically recovered by maximising the data log-likelihood, which we approach by model selection (McLachlan & Peel, 2000). However, in practice, overestimating $K$ may lead to overfitting or redundant regimes. To overcome this issue, standard selection heuristics such as the elbow method (Thorndike, 1953) can be applied. Figure 7 shows the reconstruction error for two settings: states recurrent with $N = 5$ and $K = 2$ (in red); and states determined with $N = 10$ and $K = 4$ (in blue). In both cases, the MSE plateaus when $K$ reaches its ground truth value, thus providing a principled heuristic to identify the true number of states.

### 6.2. Gene Regulatory Networks

We extend our evaluation to the inference of Gene Regulatory Networks (GRNs) during cellular differentiation, reprogramming and development, which are defined by dynamically changing gene interactions with nonstationary,

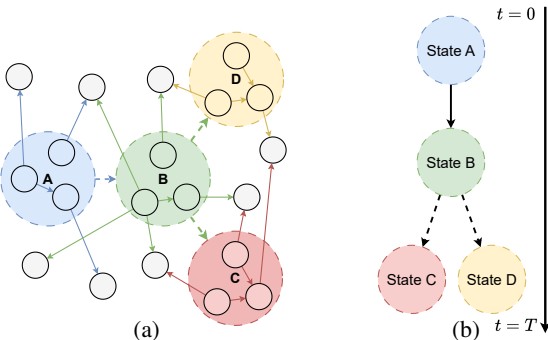

(a)          (b)

Figure 8: Example of a bifurcating network (b) from Dyn-Gen, where in (a) gray nodes denote target genes, and other colors indicate genes activated by their corresponding states.

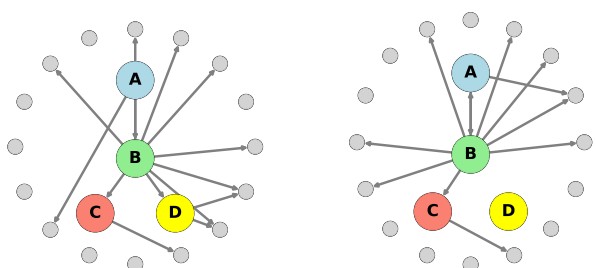

Figure 9: Conditional summary graphs extracted by SDCI. Genes are grouped by states, and gray nodes denote targets.

and nonlinear dynamics (Schiebinger et al., 2019; Yeo et al., 2021; Kamimoto et al., 2023; Bhaskar et al., 2024). We simulate gene expression data using DynGen (Cannoodt et al., 2021), a semi-synthetic simulation engine that models dynamic regulatory interactions.[2] In DynGen, an underlying state network introduces nonstationarity by activating or deactivating groups of genes, leading to time-varying structures. Fig. 8 illustrates an example of a bifurcating network, where the system transitions through different states: activating genes in $A$, then transitioning to $B$, and ending by activating genes from $C$ or $D$. See App. C.2 for details.

We simulate 1000 cells with $N = 49$ genes for $T = 50$ time steps; under a bifurcating state network with 4 regimes. We find SDCI is tailored to GRN setups, since the state variables ($K = 2$) naturally activate or deactivate gene expressions. We evaluate SDCI in the 3 scenario classes, and we leverage ground-truth information to manually activate or deactivate genes in the observed case. In Table 1 we report results in comparison to causal discovery baselines in terms of estimating the underlying GRN structure, where we observe SDCI shows superior performance in terms of $F_1$ score. In the observed case, we find that state auxiliary information does not provide a significant advantage, as

---

[2]We do not use the widely-used DREAM3 benchmark (Marbach et al., 2010) since it is limited by only simulating steady-state and individual gene perturbation conditions.

Table 1: AUROC and $F_1$ score of the GRN structure.

| METHOD | AUROC | $F_1$ SCORE |
|---|---|---|
| R-PCMCI (Saggioro et al., 2020) | 0.887 | 0.062 |
| N-GC (Tank et al., 2021) | 0.556 | 0.093 |
| ACD (Löwe et al., 2022) | 0.714 | 0.187 |
| Rhino (Gong et al., 2023) | 0.933 | 0.081 |
| SDCI – Observed | 0.805 | 0.182 |
| SDCI – Determined | **0.936** | **0.347** |
| SDCI – Recurrent | 0.855 | 0.276 |

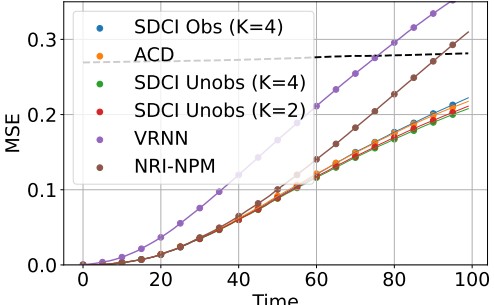

Figure 10: Test error on forecasts. The dashed line represents the mean value.

gene deactivation is typically gradual rather than abrupt. The recurrent case suffers from the underlying exponential cost of exact inference (approximated by $q_\phi$), which is critical in high variable scenarios. In contrast, the determined case proves to be a simple yet effective approach that adapts to the time-varying dynamics in GRNs. We visualise the estimated strucutre in Fig. 9, where genes are grouped by their associated states. SDCI successfully detects deactivated gene interactions, except for gene groups B and C; and thus provides a robust framework for realistic GRNs with direct applications to tasks such as cell type specific regulatory network inference.

### 6.3. NBA Player Trajectories

We consider NBA player movements (Linou, 2016), a real-world multi-agent trajectory dataset with highly nonlinear & nonstationary dynamics. See Appendix C.3 for details of the dataset, including our design for auxiliary states (as ground-truth is unavailable). We evaluate SDCI under observed and determined states. To simplify the task, we model the trajectories of the players (position and velocity) conditioned on the ball's position and velocity. This is achieved by modifying Eq. (10) to include the ball features in the message passing aggregation of the decoder. In addition to ACD, we compare against non-causal baselines VRNN (Chung et al., 2015) and NRI-NPM (Chen et al., 2021) (Appendix B.4). All the models are trained on sequences of length $T = 100$.

Fig. 10 shows the average forecast error (MSE) of the player

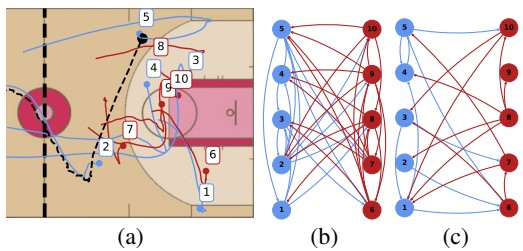

(a)       (b)       (c)

Figure 11: Illustration of learned structures on SDCI ($K = 4$) given a sample (a), where (b), (c) correspond to $\mathcal{G}_2^{CS}$ and $\mathcal{G}_4^{CS}$ respectively.

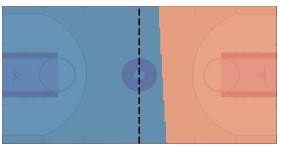
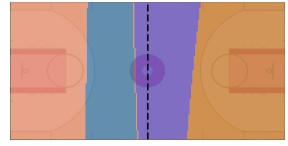

(a) $K = 2$ state values.     (b) $K = 4$ state values.

Figure 12: Learned regimes from SDCI on the NBA dataset. The colour maps refer to the state posterior distribution $q_\phi(s_t^{(i)} = k | \boldsymbol{x}_t^{(i)}), k = \{1, \ldots, K\}$.

positions on a held-out dataset. Overall GNN-based methods outperform VRNN. Moreover, ACD and SDCI achieve better long-term forecasts, despite using first-order Markov transitions. This is likely due to the "no-edge" interaction for reducing error accumulation through time. The hidden state setting allows SDCI to have a slight advantage over ACD thanks to adapting player interactions through time.

Fig. 11 visualizes the extracted conditional summary graphs from SDCI for a sample trajectory (additional visualizations in Appendix E.2). We observe that SDCI extracts interpretable structures relevant to basketball plays: Players 2 and 5 receive the majority of interactions (Fig. 11b) as Player 2 passes the ball to Player 5. Also interactions from the read team (defense) to the blue team (offense) are more prominent. Notably, SDCI adapts to nonstationary dynamics by controlling interaction sparsity and switching regimes throughout sequences.

While our identifiability theory does not require a predetermined number of states values $K$, in practice accurate estimation of $K$ remains a challenge, where different $K$ may return different results. For instance, Fig. 12 shows the state posterior as a function of player positions on the court. For $K = 2$, states change near the mid-court line, reflecting common basketball play strategies where players transition between "defense" and "offense". The $K = 4$ case returns a more fine-grained result by further partitioning the court, with boundaries positioned near the three-point line. Notably these insights arise despite model simplifications, where the states are made dependent solely on positions, excluding velocities. This highlights how SDCI can adapt

to diverse modelling choices while extracting interpretable patterns in complex, nonstationary systems.

## 7. Conclusions

We have developed SDCI for causal discovery in conditionally stationary time series. The key innovation is the new concept of conditional summary graphs that are identifiable under a broad class of underlying state dependencies Evaluations on semi-synthetic data show SDCI's superior performance in extracting the underlying causal graph and its potential for biology applications. Importantly, the improvement of SDCI over VRNN and NRI on modeling NBA player movements demonstrate the promise of causality-driven methods for forecasting and data interpretability.

## Impact Statement

This paper presents work whose goal is to advance the field of Machine Learning. There are many potential societal consequences of our work, none which we feel must be specifically highlighted here.

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

# A. Identifiability Proofs

We have introduced *conditionally stationary* time series in Section 3. Here, we will revise the model assumptions introduced in the main text, and present of our identifiability theory; starting from general Markov Switching Models (m1-m3), following with conditional summary graph identification under (m4), and finishing with identifiability of the labels of conditional effects (m5) utilised in SDCI.

## A.1. Preliminaries

We review the main model assumptions that characterise SDCI. We start with the assumptions that apply to general MSMs, where we have a single global state at each time step t. In this case we denote the global state as $u_t =: \varphi(\boldsymbol{s}_t) \in \{1, \ldots, U\}$ for some injective $\varphi : \{1, \ldots, K\}^N \to \{1, \ldots, U\}$ (we use $\boldsymbol{s}_t$ when referring to multiple states in SDCI).

**(m1) Conditional first-order Markov transitions** controlled by $u_{t-1}$ only:

$$p(\boldsymbol{x}_t|\boldsymbol{x}_{1:t-1}, \boldsymbol{u}_{1:t-1}) = p(\boldsymbol{x}_t|\boldsymbol{x}_{t-1}, u_{t-1}), \quad \forall t \in \{2, ..., T\}. \tag{21}$$

**(m2) Conditional stationarity:** the conditional transition distribution does not change during time: for any $u \in \{1, ..., U\}$ we have for any $t \neq t'$, any $\boldsymbol{\beta}, \boldsymbol{\gamma} \in \mathbb{R}^{Nd}$

$$p(\boldsymbol{x}_t = \boldsymbol{\beta}|\boldsymbol{x}_{t-1} = \boldsymbol{\gamma}, u_{t-1} = u) = p(\boldsymbol{x}_{t'} = \boldsymbol{\beta}|\boldsymbol{x}_{t'-1} = \boldsymbol{\gamma}, u_{t'-1} = u). \tag{22}$$

**(m3) Factorisation structure based on the ANM model with Gaussian noise:**

$$p(\boldsymbol{x}_t|\boldsymbol{x}_{t-1}, u_{t-1}) = \prod_{j=1}^N \mathcal{N}\left(\boldsymbol{x}_t^{(j)}; f_{u_{t-1}}^{(j)}\left(\boldsymbol{x}_{t-1}^{(i)}|\boldsymbol{x}_{t-1}^{(i)} \in \mathbf{PA}_{u_{t-1}}^{(j)}(t-1)\right), \sigma^2\mathbf{I}\right), \quad \forall t \in \{2, ..., T\}. \tag{23}$$

Again, we formalise assumptions specific to *conditionally stationary* data, where the total state configuration is denoted by grouping all the individual states $\boldsymbol{s}_t = \{s_t^{(1)}, \ldots, s_t^{(N)}\} \in \{1, \ldots, K\}^N$. Note that this defines a specific structure in a general MSM with $U := K^N$ global states.

**(m4) Multi-state dependence:** For each variable $i \in \{1, \ldots, N\}$ at time $t \in \{1, \ldots, T\}$: $\boldsymbol{x}_t^{(i)}$, there exists a state variable $s_t^{(i)} \in \{1, \ldots, K\}$, such that $\boldsymbol{s}_{t-1} := (s_{t-1}^{(1)}, \ldots, s_{t-1}^{(N)}) \in \{1, \ldots, K\}^N$. For any variable $j \in \{1, \ldots, N\}$, $\mathbf{PA}_{\boldsymbol{s}_{t-1}}^{(j)}(t-1)$, is defined as follows:

$$\mathbf{PA}_{\boldsymbol{s}_{t-1}}^{(j)}(t-1) := \left\{\boldsymbol{x}_{t-1}^{(i)}|j \in \mathcal{C}_k^{(i)}, k = s_{t-1}^{(i)}, 1 \leq i \leq N\right\}. \tag{24}$$

where $\mathcal{C}_k^{(i)} \subseteq \mathcal{V}$ denotes the *outgoing edge structure* (children) of variable $i \in \{1, \ldots, K\}$, corresponding to a state $k \in \{1, \ldots, K\}$.

**(m5) Global function-type dependence:** We allow $n_\epsilon$ functional effects (including the no-edge effect). We define the *labels of conditional effects* as

$$\mathcal{W} := \left\{w_{i,j,k} \in \{0, \ldots, n_\epsilon - 1\} : i, j \in \{1 \ldots, N\}, k \in \{1, \ldots, K\}, w_{i,j,k} = 0 \iff j \notin \mathcal{C}_k^{(i)}\right\}, \tag{25}$$

where each $w_{i,j,k}$ denotes an edge-type from variable $i$ to $j$ at state $k$, associated to variable $i$. Each edge-type corresponds to a different function over the total function interactions $\mathcal{F} = \{f_0, \ldots, f_{n_\epsilon-1}\}$, where $f_0(\cdot) = \mathbf{0}$ from Eq. (25). Given a state configuration $\boldsymbol{s}_{t-1}$, $f_{\boldsymbol{s}_{t-1}}^{(j)} : \mathbb{R}^d \to \mathbb{R}^d$ in Eq. (23) is defined as follows for any $\boldsymbol{x}_{t-1} \in \mathbb{R}^{Nd}$:

$$f_{\boldsymbol{s}_{t-1}}^{(j)}(\boldsymbol{x}_{t-1}) \quad := \quad g\left(\boldsymbol{x}_{t-1}^{(j)}, \left\{f_{e_i}\left(\boldsymbol{x}_{t-1}^{(i)}, \boldsymbol{x}_{t-1}^{(j)}\right)|i \neq j\right\}\right), \quad \text{where } e_i \quad = \quad w_{i,j,s_{t-1}^{(i)}} \quad \in \quad \{0, \ldots, n_\epsilon\}, \tag{26}$$

where $e_i, i \in \{1, \ldots, N\}$ denotes the interactions between variables $i$ and $j$, which are aggregated by a permutation invariant function $g$.

## A.2. Partial Identifiability

We wish to analyse the identifiability of the labels $\mathcal{W}$ for SDCI (m1-m5) in the case of unobserved states. However, we first require developing identifiability for regime-dependent and state-dependent structures, i.e $\mathcal{G}_{1:U}^{RD}$ and $\mathcal{G}_{1:K}^{CS}$ respecitvely. We define identifiability of the regime-dependent structures for a general MSM (m1-m3).

**Definition A.1** (Partial identifiability of regime-dependent graph). Given observations $\boldsymbol{x}_{1:T}$ and a family of models $\mathcal{M}$ satisfying (m1-m3), we say $\mathcal{M}$ is *partially identifiable with respect to its regime-dependent graph* if for any $p, \hat{p} \in \mathcal{M}$, with corresponding $\mathbf{PA}_u^{(i)}(t-1)$ and $\widehat{\mathbf{PA}}_{\hat{u}}^{(i)}(t-1)$ for any $i \in \{1, \ldots, N\}$, $u \in \{1, \ldots, U\}$, and $\hat{u} \in \{1, \ldots, \hat{U}\}$), such that $p(\boldsymbol{x}_{1:T}) = \hat{p}(\boldsymbol{x}_{1:T})$; $U = \hat{U}$ and there exists a permutation $\tau \in S_U$ such that $\mathbf{PA}_u^{(i)}(t-1) = \widehat{\mathbf{PA}}_{\tau(u)}^{(i)}(t-1)$ for $i \in \{1, \ldots, N\}$, and $u \in \{1, \ldots, U\}$.

We also define the identifiability of the outgoing edges $C_k^{(i)}$ in terms of $\mathcal{G}_{1:K}^{CS}$, given a model with $N$ state variables and multi-state dependence (m1-m4).

**Definition A.2** (Partial identifiability of outgoing edge structure). Given observations $\boldsymbol{x}_{1:T}$ and a family of models $\mathcal{M}$ satisfying (m1-m4), we say $\mathcal{M}$ is *partially identifiable with respect to the outgoing edge structure* if for any $p, \hat{p} \in \mathcal{M}$, with corresponding $\mathcal{C}_k^{(i)}$ and $\widehat{\mathcal{C}}_{\hat{k}}^{(i)}$ for any $i \in \{1, \ldots, N\}$, $k \in \{1, \ldots, K\}$, and $\hat{k} \in \{1, \ldots, \hat{K}\}$), such that $p(\boldsymbol{x}_{1:T}) = \hat{p}(\boldsymbol{x}_{1:T})$; $K = \hat{K}$ and there exist permutations $\sigma^{(i)} \in S_K$ such that $\mathcal{C}_k^{(i)} = \hat{\mathcal{C}}_{\sigma^{(i)}(k)}^{(i)}$ for $i \in \{1, \ldots, N\}$, and $k \in \{1, \ldots, K\}$.

Finally, we further define identifiability of the labels of conditional effects under global function-type dependence (m5).

**Definition A.3** (Partial identifiability of conditional effects). Given observations $\boldsymbol{x}_{1:T}$ and a family of models $\mathcal{M}$ satisfying (m1-m5), we say $\mathcal{M}$ is *partially identifiable with respect to the labels of conditional effects* if for any $p, \hat{p} \in \mathcal{M}$, with corresponding $\mathcal{W}$ and $\widehat{\mathcal{W}}$ such that $p(\boldsymbol{x}_{1:T}) = \hat{p}(\boldsymbol{x}_{1:T})$; $K = \hat{K}$ and there exist permutations $\pi \in S_{n_\epsilon}$ and $\sigma^{(i)} \in S_K$ such that for any $i, j \in \{1, \ldots, N\}$ and $k \in \{1, \ldots, K\}$:

$$w_{i,j,k} = \hat{w}_{i,j,\sigma^{(i)}(k)} \iff w_{i,j,k} = 0, \tag{27}$$

$$w_{i,j,k} = \pi\left(\hat{w}_{i,j,\sigma^{(i)}(k)}\right) \iff w_{i,j,k} \neq 0. \tag{28}$$

As mentioned in the main text, these definition consider identifiability is *partial* since it does not cover the structure capturing interaction between $\boldsymbol{x}_{1:T}$ and $\boldsymbol{s}_{1:T}$ as well as those within $\boldsymbol{s}_{1:T}$.

## A.3. Sufficiency Conditions for Identifiability

The defined partial graph identifiability cannot be achieved without further assumptions for (m1-m5). Below we list the sufficient conditions for such identifiability, some of them which we already introduced in the main text.

**(a1) Unique indexing of outgoing structure.** Here we assume that for each state $s_{t-1}^{(i)} \in \{1, ..., K\}^N$, $i \in \{1, \ldots, N\}$ the underlying graph representing the direct causes from variable $i$ to the rest of the variables is unique.

$$\forall k, k' \in \{1, ..., K\}, k \neq k' \quad \Leftrightarrow \quad \mathcal{C}_k^{(i)} \neq \mathcal{C}_{k'}^{(i)}, \quad \forall i \in \{1, ..., N\}. \tag{29}$$

This assumption has equivalences to MSMs, i.e. when considering all state variables, $\boldsymbol{s}_{t-1} \in \{1, \ldots, K\}^N$ as a global state, $u_{t-1} \in \{1, \ldots, K^N\}$.

**(b1) Unique indexing of regime-dependent graph structure.** For each possible state $u_{t-1} \in \{1, \ldots, U\}$ the underlying graph representing the direct causes between variables is unique. In other words, the resulting $U$ graphs are different. This implies that the following holds:

$$\forall u, u' \in \{1, ..., U\}, u \neq u' \quad \Leftrightarrow \quad \exists j \in \{1, ..., N\} \text{ s.t. } \mathbf{PA}_{u_{t-1}=u}^{(j)}(t-1) \neq \mathbf{PA}_{u_{t-1}=u'}^{(j)}(t-1). \tag{30}$$

Assume an invertible injective mapping $\varphi : \{1, \ldots, K\}^N \to \{1, \ldots, K^N\}$ such that we can map a specific state configuration $(k_1, \ldots, k_N)$ to an assigned value $k_{1:N} \in \{1, \ldots, K^N\}$ that is, $k_{1:N} = \varphi(k_1, \ldots, k_N)$. With this view, we can write the state variables in terms of one global state variable with $K^N$ regimes. We connect assumption (a1) to assumption (b1) defined for MSMs as follows.

**Proposition A.4.** *Assumption (a1) implies assumption (b1).*

*Proof.* Recall the definition of $\mathbf{PA}^{(j)}_{s_{t-1}}$:

$$\mathbf{PA}^{(j)}_{s_{t-1}}(t-1) := \{\boldsymbol{x}^{(i)}_{t-1} \mid j \in \mathcal{C}^{(i)}_k, k = s^{(i)}_{t-1}, 1 \le i \le N\}. \tag{31}$$

Given two state configurations $(k_1, \ldots, k_N), (k'_1, \ldots, k'_N) \in \{1, \ldots, K\}^N$ that differ in at least one $k_i, i \in \{1, \ldots, N\}$, under (a1), we have $\mathcal{C}^{(i)}_{k_i} \ne \mathcal{C}^{(i)}_{k'_i}$. Again from (a1), $\exists j \in \{1, \ldots, N\}$ such that $j \in \mathcal{C}^{(i)}_{k_i}$ and $j \notin \mathcal{C}^{(i)}_{k'_i}$. From Eq. (31), and by writing $\mathbf{PA}^{(j)}_{s_{t-1}}(t-1)$, as a MSM with one global state ($u_t := \boldsymbol{s}_t$) we have

$$\exists j \in \{1, \ldots, N\} \; s.t. \; \boldsymbol{x}^{(i)}_{t-1} \in \mathbf{PA}^{(j)}_{u_{t-1}=\varphi(k_1,\ldots,k_N)}(t-1), \text{ and } \boldsymbol{x}^{(i)}_{t-1} \notin \mathbf{PA}^{(j)}_{u_{t-1}=\varphi(k'_1,\ldots,k'_N)}(t-1)$$

$$\implies \exists j \in \{1, \ldots, N\} \; s.t. \; \mathbf{PA}^{(j)}_{u_{t-1}=\varphi(k_1,\ldots,k_N)}(t-1) \ne \mathbf{PA}^{(j)}_{u_{t-1}=\varphi(k'_1,\ldots,k'_N)}(t-1) \tag{32}$$

Therefore, assumption (b1) also holds. $\qquad\square$

We now revisit the functional assumptions introduced in the main text, and establish connections to MSMs.

**(a2) Unique indexing of function types.** Under the model assumptions (m1-m5), we assume the following:

**(a2.1):** For any $j \in \{1, \ldots, N\}$, and given $g(\boldsymbol{x}^{(j)}, \{\boldsymbol{h}^{(i,j)}|i \ne j\})$, where $\boldsymbol{h}^{(i,j)} = f_e(\boldsymbol{x}^{(i)}, \boldsymbol{x}^{(j)})$ for some $e \in \{0, \ldots, n_\epsilon - 1\}$, the partial derivative $\frac{\partial g}{\partial \boldsymbol{h}^{(i,j)}}$ is non-zero almost everywhere for any $i \in \{1, \ldots, N\}$.

**(a2.2):** Edge-types differ almost everywhere: $f_0 := \boldsymbol{0}$, and for $e \ne e', \in \{0, n_\epsilon - 1\}$, the following set has zero measure:

$$\mathcal{X}_{e,e'} := \left\{ \boldsymbol{x}_1 \in \mathbb{R}^d : \exists \boldsymbol{x}_2 \in \mathbb{R}^d, \quad \frac{\partial f_e(\boldsymbol{x}_1, \boldsymbol{x}_2)}{\partial \boldsymbol{x}_1} = \frac{\partial f'_e(\boldsymbol{x}_1, \boldsymbol{x}_2)}{\partial \boldsymbol{x}_1} \right\}. \tag{33}$$

**(b2) Functional faithfulness.** This condition considers the functional properties of $f^{(j)}_{s_{t-1}}$ in terms of its faithfulness to the graph structure. We require the following sense of faithfulness in terms of the functional behaviour for all $\mathbf{s}_{t-1} \in \{1, ..., K\}^K$:

(b2.1) $f^{(j)}_{s_{t-1}}$ is differentiable w.r.t. $\mathbf{PA}^{(j)}_{s_{t-1}}(t-1)$ almost everywhere. Also all the entries of the Jacobian matrix $\frac{df^{(j)}_{s_{t-1}}}{d\mathbf{PA}^{(j)}_{s_{t-1}}(t-1)}$, when well defined, are non-zero almost everywhere w.r.t. $\mathbf{PA}^{(j)}_{s_{t-1}}(t-1)$.

(b2.2) If $\boldsymbol{x}^{(i)}_{t-1} \notin \mathbf{PA}^{(j)}_{s_{t-1}}(t-1)$, then $f^{(j)}_{s_{t-1}}$ is a constant w.r.t. $\boldsymbol{x}^{(i)}_{t-1}$.

Intuitively, this faithfulness condition requires the output of the function $f^{(j)}_{s_{t-1}}$ to vary if and only if $\mathbf{PA}^{(j)}_{s_{t-1}}(t-1)$ varies. We also connect (a2) to (b2).

**Proposition A.5.** *(a2) implies (b2).*

To see this, note that (a2.1) and (a2.2) force the derivatives w.r.t $\mathbf{PA}^{(j)}_{s_{t-1}}(t-1)$ to be non-zero almost everywhere (which implies (b2.1). Furthermore, this is only violated in a zero measure set of points, or when variable $i$ interacts with $j$ via $f_0$, but this only is possible when $\boldsymbol{x}^{(i)}_{t-1} \notin \mathbf{PA}^{(j)}_{s_{t-1}}(t-1)$, which implies (b2.2).

### A.4. Identifiability of Markov Switching Models

With the above assumptions we can now state the following identifiability results:

**Theorem A.6.** *A Markov Switching Model (m1-m3) under assumptions (b1) and (b2) is partially identifiable with respect to its regime-dependent graph (Def. A.1).*

*Proof.* Assume there exist two MSMs satisfying $p(\boldsymbol{x}_{1:T}) = \hat{p}(\boldsymbol{x}_{1:T})$. Since wlog. $p(\boldsymbol{x}_{1:t}) = p(\boldsymbol{x}_t|\boldsymbol{x}_{1:t-1})p(\boldsymbol{x}_{1:t-1})$, the fact that $p(\boldsymbol{x}_{1:T}) = \hat{p}(\boldsymbol{x}_{1:T})$, and $p(\boldsymbol{x}_t|\boldsymbol{x}_{1:t-1})$ is a probability distribution imply that

$$p(\boldsymbol{x}_t|\boldsymbol{x}_{1:t-1}) = \hat{p}(\boldsymbol{x}_t|\boldsymbol{x}_{1:t-1}), \quad \forall t = 2, ..., T. \tag{34}$$

Now since the model assumes (m1) a conditional first-order Markov structure controlled by the previous state $s_{t-1}$ (Eq. (21)), we can show that the conditional distribution is a finite mixture distribution:

$$p(\boldsymbol{x}_t|\boldsymbol{x}_{1:t-1}) = \sum_{u=1}^{U} p(u_{t-1} = u|\boldsymbol{x}_{1:t-1})p(\boldsymbol{x}_t|\boldsymbol{x}_{t-1}, u_{t-1} = u). \tag{35}$$

We assume (b1) and functional faithfulness (b2). Since (m3) $p(\boldsymbol{x}_{1:t}|\boldsymbol{x}_{t-1}, u_{t-1} = u)$ is Gaussian (Eq. (23)), using the identifiability result of Gaussian finite mixture (Yakowitz & Spragins, 1968) we have $U = \hat{U}$, and for almost any given $\boldsymbol{x}_{t-1} = \boldsymbol{\alpha} \in \mathbb{R}^{Nd}$ (modulo some zero-measure sets) we have the following result: for any $u \in \{1, ..., U\}$, there exists $\hat{u}(\boldsymbol{\alpha}, u) \in \{1, ..., U\}$ such that

$$\begin{aligned} p(u_{t-1} = u|\boldsymbol{x}_{1:t-2}, \boldsymbol{x}_{t-1} = \boldsymbol{\alpha}) &= \hat{p}(u_{t-1} = \hat{u}(\boldsymbol{\alpha}, u)|\boldsymbol{x}_{1:t-2}, \boldsymbol{x}_{t-1} = \boldsymbol{\alpha}), \\ p(\boldsymbol{x}_t|\boldsymbol{x}_{t-1} = \boldsymbol{\alpha}, u_{t-1} = u) &= \hat{p}(\boldsymbol{x}_t|\boldsymbol{x}_{t-1} = \boldsymbol{\alpha}, u_{t-1} = \hat{u}(\boldsymbol{\alpha}, u)). \end{aligned} \tag{36}$$

To clarify, Proposition 2 in Yakowitz & Spragins (1968) establishes multivariate Gaussian distributions are identifiable if and only if the means or covariances are distinct. Our assumptions (b1) and (b2) ensure distinct means for almost any $\boldsymbol{x}_{t-1} = \boldsymbol{\alpha} \in \mathbb{R}^{Nd}$.

The (m3) factorised Gaussian structure (Eq. (23)) and the above identification result further indicate that

$$p(\boldsymbol{x}_t^{(j)}|\boldsymbol{x}_{t-1} = \boldsymbol{\alpha}, u_{t-1} = u) = \hat{p}(\boldsymbol{x}_t^{(j)}|\boldsymbol{x}_{t-1} = \boldsymbol{\alpha}, u_{t-1} = \hat{u}(\boldsymbol{\alpha}, u)), \quad \forall j = 1, ..., N. \tag{37}$$

Again wlog., let us denote the mean function of $p(\boldsymbol{x}_t^{(j)}|\boldsymbol{x}_{t-1} = \boldsymbol{\alpha}, u_{t-1} = u)$ as $f_u^{(j)}$ for $j = 1, ..., N$ and $u = 1, ..., K$, and we also use a short notation $\mathbf{PA}_u^{(j)}(t-1)$ to denote the input variables for $f_u^{(j)}$. Under (b2.1) of the (b2) functional faithfulness assumption, and given that $U, N < +\infty$, there exist an open set $\mathcal{X} \subset \mathbb{R}^{Nd}$ such that $\mu(\mathcal{X}) = \mu(\mathbb{R}^{Nd})$, where $\mu(\cdot)$ denotes the Lebesgue measure of a Euclidean space, and both Eq. (37) and the following condition hold (note that $\mathbf{PA}_u^{(j)}(t-1) \subset \boldsymbol{x}_{t-1}$):

$$\text{all the entries of } \frac{df_u^{(j)}}{d\mathbf{PA}_u^{(j)}(t-1)}|_{\boldsymbol{x}_{t-1} = \boldsymbol{\alpha}} \text{ are non-zero}, \quad \forall \boldsymbol{\alpha} \in \mathcal{X}, \ \forall j \in \{1, ..., N\}. \tag{38}$$

To see this: under (b2.1) we have for each $u \in \{1, ..., U\}$ there exists an open set $\mathcal{X}_u \subset \mathbf{PA}_u^{(j)}(t-1)$ with $\mu(\mathcal{X}_u) = \mu(\mathbf{PA}_u^{(j)}(t-1))$ such that the partial derivatives computed within this set are non-zero. Then we can construct $\mathcal{X}$ as follows:

$$\mathcal{X} = \check{\mathcal{X}} \cap \hat{\mathcal{X}}, \quad \check{\mathcal{X}} = \bigcap_{u=1}^{U} (\mathcal{X}_u \times \{\boldsymbol{x}_{t-1}^{(i)} \in \mathbb{R}^d | \boldsymbol{x}_{t-1}^{(i)} \notin \mathbf{PA}_u^{(j)}(t-1)\}),$$

$$\hat{\mathcal{X}} = \bigcap_{u=1}^{U} (\hat{\mathcal{X}}_u \times \{\boldsymbol{x}_{t-1}^{(i)} \in \mathbb{R}^d | \boldsymbol{x}_{t-1}^{(i)} \notin \widehat{\mathbf{PA}}_u^{(j)}(t-1)\}). \tag{39}$$

We can show wlog. that $\mu(\mathcal{X}_u \times \{\boldsymbol{x}_{t-1}^{(i)} \in \mathbb{R}^d | \boldsymbol{x}_{t-1}^{(i)} \notin \mathbf{PA}_u^{(j)}(t-1)\}) = \mu(\mathbb{R}^{Nd})$ since the Lebesgue measure of $\mathbb{R}^{Nd}$ is a product measure. Therefore using the union bound we also have $\mu(\mathcal{X}) = \mu(\mathbb{R}^{Nd})$.

Now we show that $\hat{u}(\boldsymbol{\alpha}, u)$ is a constant for $\boldsymbol{\alpha} \in \mathcal{X}$ almost everywhere, and those $\boldsymbol{\alpha}$ values satisfy $\hat{u}(\boldsymbol{\alpha}, u) = \hat{u}(u)$ and $\mathbf{PA}_u^{(j)}(t-1) = \widehat{\mathbf{PA}}_{\hat{u}(u)}^{(j)}(t-1)$ for all $j \in \{1, ..., N\}$.

(i) By model definition (m3) (Eq. (23)) and assumption (b2.1), within $\boldsymbol{x}_{t-1} = \boldsymbol{\alpha} \in \mathcal{X}$ we have $p(\boldsymbol{x}_t^{(j)}|\boldsymbol{x}_{t-1}, u_{t-1} = u) = p(\boldsymbol{x}_t^{(j)}|\mathbf{PA}_u^{(j)}(t-1), u_{t-1} = u)$. Similarly for $\hat{p}$ we have $\hat{p}(\boldsymbol{x}_t^{(j)}|\boldsymbol{x}_{t-1}, u_{t-1} = \hat{u}(\boldsymbol{\alpha}, u)) = \hat{p}(\boldsymbol{x}_t^{(j)}|\widehat{\mathbf{PA}}_{\hat{u}(\boldsymbol{\alpha}, u)}^{(j)}(t-1), u_{t-1} = \hat{u}(\boldsymbol{\alpha}, u))$.

Then from Eq. (37) and assumption (b2.2), there exist at most a zero-measure set $\mathcal{X}_0 \subset \mathcal{X}$ of $\boldsymbol{\alpha}$ values such that $\widehat{\mathbf{PA}}_{\hat{u}(\boldsymbol{\alpha},u)}^{(j)}(t-1)) \neq \mathbf{PA}_u^{(j)}(t-1)$. Otherwise, we can show that there exist two $\boldsymbol{\alpha} \neq \boldsymbol{\alpha}' \in \mathcal{X}_0$ with $\widehat{\mathbf{PA}}_{\hat{u}(\boldsymbol{\alpha},u)}^{(j)}(t-1)) = \widehat{\mathbf{PA}}_{\hat{u}(\boldsymbol{\alpha}',u)}^{(j)}(t-1)) \neq \mathbf{PA}_u^{(j)}(t-1)$, which contradicts with Eq. (37) and (b2.2). This means, by Eq. (23) again, and notice that $\mu(\mathcal{X}\backslash\mathcal{X}_0) = \mu(\mathcal{X})$:

$$\widehat{\mathbf{PA}}_{\hat{u}(\boldsymbol{\alpha},u)}^{(j)}(t-1) = \mathbf{PA}_u^{(j)}(t-1), \quad \forall \boldsymbol{\alpha} \in \mathcal{X}\backslash\mathcal{X}_0, \ \forall j \in \{1,...,N\}.$$

(ii) Under (b1) unique indexing of causal graph structure, we have $\hat{u}(\boldsymbol{\alpha}, u)$ as a constant w.r.t. $\boldsymbol{\alpha} \in \mathcal{X}\backslash\mathcal{X}_0$. To see this, if there exist $\boldsymbol{\alpha}, \boldsymbol{\alpha}' \in \mathcal{X}\backslash\mathcal{X}_0$ such that $\hat{u}(\boldsymbol{\alpha}, u) \neq \hat{u}(\boldsymbol{\alpha}', u)$, then from (b1), there exists $j \in \{1, ..., N\}$ such that $\widehat{\mathbf{PA}}_{\hat{u}(\boldsymbol{\alpha},u)}^{(j)}(t-1)) \neq \widehat{\mathbf{PA}}_{\hat{u}(\boldsymbol{\alpha}',u)}^{(j)}(t-1))$, a contradiction to point (i) above. We now write $\hat{u}(\boldsymbol{\alpha}, u) = \hat{u}(u)$ w.l.o.g., then we have

$$\mathbf{PA}_u^{(j)}(t-1) = \widehat{\mathbf{PA}}_{\hat{u}(u)}^{(j)}(t-1), \quad \forall \boldsymbol{\alpha} \in \mathcal{X}\backslash\mathcal{X}_0, \ \forall j \in \{1,...,N\}.$$

In summary, we have shown that there exists a permutation $\tau \in S_U$ such that $\hat{u} = \tau(u)$ and the following equivalence holds for all $j \in \{1, ..., N\}$ and almost everywhere for $\boldsymbol{x}_{t-1} \in \mathbb{R}^{Nd}$:

$$p(\boldsymbol{x}_t^{(j)}|\boldsymbol{x}_{t-1}, u_{t-1} = u) = \hat{p}(\boldsymbol{x}_t^{(j)}|\boldsymbol{x}_{t-1}, u_{t-1} = \tau(u)), \quad \mathbf{PA}_u^{(j)}(t-1) = \widehat{\mathbf{PA}}_{\tau(u)}^{(j)}(t-1),$$
$$\forall j \in \{1, ..., N\}, \quad \forall \boldsymbol{x}_{t-1} \in \mathcal{X}\backslash\mathcal{X}_0, \quad \mu(\mathcal{X}\backslash\mathcal{X}_0) = \mu(\mathbb{R}^{Nd}). \tag{40}$$

$\square$

## A.5. Identifiability of Conditionally Stationary Time Series

We start by introducing assumption (m4) and establish identifiability of the conditional summary graph under unique indexing of outgoing edges (a1) and functional faithfulness (b1).

**Theorem A.7.** *The multi-state dependent model (m1-m4) is partially identifiable w.r.t. the outgoing edge structure (Def. A.2) up to permutation equivalence of the states, if it satisfies the assumptions of (a1) unique indexing of causal graph structure and (b2) functional faithfulness.*

*Proof.* Assume there exist two models under (m1-m4), satisfying $p(\boldsymbol{x}_{1:T}) = \hat{p}(\boldsymbol{x}_{1:T})$. For simplicity, we first want to view the above as a mixture model of $K^N$ regimes with one global state. Assume again an invertible injective mapping $\varphi : \{1, \ldots, K\}^N \to \{1, \ldots, K^N\}$ such that we can map a specific state configuration $(k_1, \ldots, k_N)$ to an assigned value $k_{1:N} \in \{1, \ldots, K^N\}$ that is, $k_{1:N} = \varphi(k_1, \ldots, k_N)$. Therefore, for any $t \in \{1, \ldots, T\}$, we can obtain $u_t = \varphi(\boldsymbol{s}_t)$ to reduce the multi-state model into a general MSM. Given that assumption (a1) implies (b1), from Theorem A.6 there exists a permutation permutation $\tau \in S_{K^N}$ such that $\hat{k}_{1:N} = \tau(k_{1:N})$ and the following equivalence holds for all $j \in \{1, ..., N\}$ and almost everywhere for $\boldsymbol{x}_{t-1} \in \mathbb{R}^{Nd}$:

$$p(\boldsymbol{x}_t^{(j)}|\boldsymbol{x}_{t-1}, u_{t-1} = k_{1:N}) = \hat{p}(\boldsymbol{x}_t^{(j)}|\boldsymbol{x}_{t-1}, u_{t-1} = \tau(k_{1:N})), \quad \mathbf{PA}_{k_{1:N}}^{(j)}(t-1) = \widehat{\mathbf{PA}}_{\tau(k_{1:N})}^{(j)}(t-1),$$
$$\forall j \in \{1, ..., N\}, \quad \forall \boldsymbol{x}_{t-1} \in \mathcal{X}\backslash\mathcal{X}_0, \quad \mu(\mathcal{X}\backslash\mathcal{X}_0) = \mu(\mathbb{R}^{Nd}). \tag{41}$$

We can revert the above back to the multi-state model notation (m1-m4), obtaining the following equivalence for the causal effects:

$$\mathbf{PA}_{(k_1,...,k_N)}^{(j)}(t-1) = \widehat{\mathbf{PA}}_{(\varphi \circ \tau \circ \varphi^{-1})(k_1,...,k_N)}^{(j)}(t-1), \tag{42}$$

where the permutation $\tau$ is general and can permute both state indices and values. From (m4), recall the definition of $\mathbf{PA}_{(k_1,...,k_N)}^{(j)}(t-1)$ for some $(k_1, \ldots, k_N) \in \{1, \ldots, K\}^N$, and consider the equivalence on $\widehat{\mathbf{PA}}_{(\hat{k}_1,...,\hat{k}_N)}^{(j)}(t-1)$ for some $(\hat{k}_1, \ldots, \hat{k}_N) \in \{1, \ldots, K\}^N$:

$$\mathbf{PA}_{\boldsymbol{s}_{t-1}=(k_1,...,k_N)}^{(j)}(t-1) = \bigcup_{i=1}^N \left\{ x_{t-1}^{(i)} \mid j \in \mathcal{C}_{k_i}^{(i)} \right\} = \bigcup_{i=1}^N \left\{ x_{t-1}^{(i)} \mid j \in \widehat{\mathcal{C}}_{\hat{k}_i}^{(i)} \right\} = \widehat{\mathbf{PA}}_{(\hat{k}_1,...,\hat{k}_N)}^{(j)}(t-1). \tag{43}$$

Then for a given $i \in \{1, \ldots, N\}$ and by fixing $k = k_i \in \{1, \ldots, K\}$, we can recover the outgoing edge structure $\mathcal{C}_k^{(i)}$ from $\mathbf{PA}_{(k_1, \ldots, k_N)}^{(j)}(t-1)$ for all $j \in \{1, \ldots, N\}$. Equivalently, by fixing $\hat{k} = \hat{k}_i \in \{1, \ldots, K\}$, we can recover $\widehat{\mathcal{C}}_{\hat{k}}^{(i)}$ from $\widehat{\mathbf{PA}}_{(\hat{k}_1, \ldots, \hat{k}_N)}^{(j)}(t-1)$ for all $j \in \{1, \ldots, N\}$

$$\mathcal{C}_k^{(i)} = \bigcup_{j=1}^{N} \left\{ j \mid x_{t-1}^{(i)} \in \mathbf{PA}_{(k_1, \ldots, k_{i-1}, k, k_{i+1}, \ldots, k_N)}^{(j)}(t-1) \right\} = \bigcup_{j=1}^{N} \left\{ j \mid x_{t-1}^{(i)} \in \widehat{\mathbf{PA}}_{(\hat{k}_1, \ldots, \hat{k}_{i-1}, \hat{k}, \hat{k}_{i+1}, \ldots, \hat{k}_N)}^{(j)}(t-1) \right\} = \widehat{\mathcal{C}}_{\hat{k}}^{(i)}.$$

Given the unique indexing assumption (a1), every $k$ necessarily needs to map to a different $\hat{k}$. Therefore, $\hat{k}$ is obtained from a permutation of $k$, which might differ among variables. Therefore, there exists $\sigma^{(i)} \in S_K$ such that $\mathcal{C}_k^{(i)} = \widehat{\mathcal{C}}_{\sigma^{(i)}(k)}^{(i)}$ for all $i \in \{1, \ldots, N\}$ and $k \in \{1, \ldots, K\}$. This implies that the multi-state dependent model (m1-m4) is partially identifiable with respect to its outgoing edge structure. $\square$

**Corollary A.8.** *Conditionally stationary time series (m1-m5) are partially identifiable w.r.t. the labels of conditional effects (Def. A.3) up to permutations, if they satisfy: (a1) unique indexing of outgoing structure, and (a2) unique indexing of function types.*

*Proof.* Under (m1-m4) and assuming (a1-a2), from Theorem A.6 the following equivalence holds for all $j \in \{1, \ldots, N\}$ and almost everywhere for $\boldsymbol{x}_{t-1} \in \mathbb{R}^{Nd}$:

$$p\left(\boldsymbol{x}_t^{(j)} | \boldsymbol{x}_{t-1}, \boldsymbol{s}_{t-1} = (k_1, \ldots, k_N)\right) = \hat{p}\left(\boldsymbol{x}_t^{(j)} | \boldsymbol{x}_{t-1}, \boldsymbol{s}_{t-1} = \left(\sigma^{(1)}(k_1), \ldots, \sigma^{(N)}(k_N)\right)\right),$$
$$\forall j \in \{1, \ldots, N\}, \quad \forall \boldsymbol{x}_{t-1} \in \mathcal{X} \backslash \mathcal{X}_0, \quad \mu(\mathcal{X} \backslash \mathcal{X}_0) = \mu(\mathbb{R}^{Nd}). \tag{44}$$

From Gaussian identifiability (Yakowitz & Spragins, 1968), we have $f_{\boldsymbol{s}_{t-1}}^{(j)}(\boldsymbol{x}_{t-1}) = \hat{f}_{\hat{\boldsymbol{s}}_{t-1}}^{(j)}(\boldsymbol{x}_{t-1})$, for each $j \in \{1, \ldots, N\}$ and any $\boldsymbol{x}_{t-1} \in \mathcal{X} \backslash \mathcal{X}_0$. From (m5), we have

$$f_{\boldsymbol{s}_{t-1}}^{(j)}(\boldsymbol{x}_{t-1}) = g\left(f_{e_{t-1}^{(1,j)}}\left(\boldsymbol{x}_{t-1}^{(1)}, \boldsymbol{x}_{t-1}^{(j)}\right), \ldots, f_{e_{t-1}^{(N,j)}}\left(\boldsymbol{x}_{t-1}^{(N)}, \boldsymbol{x}_{t-1}^{(j)}\right)\right) =$$
$$\hat{g}\left(\hat{f}_{\hat{e}_{t-1}^{(1,j)}}\left(\boldsymbol{x}_{t-1}^{(1)}, \boldsymbol{x}_{t-1}^{(j)}\right), \ldots, \hat{f}_{\hat{e}_{t-1}^{(N,j)}}\left(\boldsymbol{x}_{t-1}^{(N)}, \boldsymbol{x}_{t-1}^{(j)}\right)\right) = \hat{f}_{\hat{\boldsymbol{s}}_{t-1}}^{(j)}(\boldsymbol{x}_{t-1}), \tag{45}$$

where $e_{t-1}^{(i,j)} := w_{i,j,s_{t-1}^{(i)}}$. We can directly establish a relation from $e_{t-1}^{(i,j)}$ to $\hat{e}_{t-1}^{(i,j)}$ using the partial derivative on the mean of $\boldsymbol{x}_t^{(j)}$ with respect to $\boldsymbol{x}_{t-1}^{(i)}$

$$\frac{\partial f_{\boldsymbol{s}_{t-1}}^{(j)}(\boldsymbol{x}_{t-1})}{\partial \boldsymbol{x}_{t-1}^{(i)}} = \frac{\partial g}{\partial \boldsymbol{h}^{(i,j)}} \frac{\partial f_{e_{t-1}^{(i,j)}}}{\partial \boldsymbol{x}_{t-1}^{(i)}} = \frac{\partial \hat{g}}{\partial \hat{\boldsymbol{h}}^{(i,j)}} \frac{\partial \hat{f}_{\hat{e}_{t-1}^{(i,j)}}}{\partial \boldsymbol{x}_{t-1}^{(i)}} = \frac{\partial \hat{f}_{\hat{\boldsymbol{s}}_{t-1}}^{(j)}(\boldsymbol{x}_{t-1})}{\partial \boldsymbol{x}_{t-1}^{(i)}}, \tag{46}$$

where the derivative of $g$ is independent of $\boldsymbol{s}_{t-1}$, and also notably $j$ (which will be of great importance later). Under assumption (a2.2), the set

$$\mathcal{X}_{func} = \bigcup_{e \neq e'} \left(\mathcal{X}_{e,e'} \cup \hat{\mathcal{X}}_{e,e'}\right)$$

is zero measured. $\hat{\mathcal{X}}_{e,e'}$ denote the sets in (a2.2) for another model $\hat{p}$ under (m1-m5) that satisfies (a1-a2). Assume $\mathcal{X}_0$ contains the points where the partial derivative of $g$ with respect to $\boldsymbol{h}^{(i,j)}$ is zero, where we know it has measure zero from (a2.1). Therefore, for any $\boldsymbol{x}_0 \in \mathcal{X} \backslash (\mathcal{X}_0 \cup (\times^N \mathcal{X}_{func}))$,

$$\frac{\partial g}{\partial \boldsymbol{h}^{(i,j)}} \frac{\partial f_{e_{t-1}^{(i,j)}}}{\partial \boldsymbol{x}_{t-1}^{(i)}} \bigg|_{\boldsymbol{x}_{t-1}^{(i)} = \boldsymbol{x}_0^{(i)}} = \frac{\partial \hat{g}}{\partial \hat{\boldsymbol{h}}^{(i,j)}} \frac{\partial \hat{f}_{\hat{e}_{t-1}^{(i,j)}}}{\partial \boldsymbol{x}_{t-1}^{(i)}} \bigg|_{\boldsymbol{x}_{t-1}^{(i)} = \boldsymbol{x}_0^{(i)}}. \tag{47}$$

Note $\mathcal{X} \backslash (\mathcal{X}_0 \cup (\times^N \mathcal{X}_{func}))$ is full measured. Given that $e_{t-1}^{(i,j)} \in \{0, \ldots, n_\epsilon - 1\}$, from (a2.1-a2.2) we know that if $e_{t-1}^{(i,j)} = 0$ we have $\hat{e}_{t-1}^{(i,j)} = 0$ due to having non-zero derivatives on $\hat{f}_e$ for any $e \neq 0$. Furthermore from (a2.2), for any $e \neq e'$ the derivatives of $f_e$ and $f_{e'}$ are not equal. Then, any $e_{t-1}^{(i,j)} \neq 0$ must map uniquely to another $\hat{e}_{t-1}^{(i,j)} \neq 0$, otherwise under

assumption (a2.1-a2.2) we have a contradiction with Eq. (47), as it will imply a repetition of edge-types. Therefore for each $i$ and each $j$, there exists a permutation $\pi^{(i,j)} \in S_{n_\epsilon - 1}$ such that for any $\hat{e}_{t-1}^{(i,j)} = \pi^{(i,j)}(e_{t-1}^{(i,j)})$ for $e_{t-1}^{(i,j)} \in \{1, \ldots, n_\epsilon - 1\}$. This does not imply any direct identifiability on the set of functions $f_1, \ldots, f_{n_\epsilon - 1}$, but the labels of conditional effects $\mathcal{W}$. Due to permutation invariance of the aggregation $g$, the partial derivative terms $\frac{\partial g}{\partial h^{(i,j)}}$ are equal across $i$ and $j$ for fixed $x_0 \in \mathcal{X} \backslash (\mathcal{X}_0 \cup (\times^N \mathcal{X}_{func}))$. This forces the permutations $\pi^{(i,j)}$ to be equal, as otherwise, edge-types where $e_{t-1}^{(i,j)} \neq 0$ mapping to different $\hat{e}_{t-1}^{(i,j)}$ will violate Eq. (47) under assumptions (a2.1-a2.2). Therefore, there exists $\pi \in S_{n_\epsilon - 1}$ such that for any $\hat{e}_{t-1}^{(i,j)} = \pi(e_{t-1}^{(i,j)})$ for $e_{t-1}^{(i,j)} \in \{1, \ldots, n_\epsilon - 1\}$. Now, recall the definition of $\mathcal{W}$:

$$\mathcal{W} := \left\{ w_{i,j,k} \in \{0, \ldots, n_\epsilon - 1\} : i, j \in \{1, \ldots, N\}, k \in \{1, \ldots, K\}, w_{i,j,k} = 0 \iff j \notin \mathcal{C}_k^{(i)} \right\}. \tag{48}$$

From Theorem A.7, indexing w.r.t $k$ in the above equation is equivalent up to a permutation $\sigma^{(i)}$ on $\widehat{\mathcal{W}}$. Considering $e_{t-1}^{(i,j)} := w_{i,j,s_{t-1}^{(i)}}$, we have the following equivalence for $\widehat{\mathcal{W}}$ when $w_{i,j,k} \neq 0$, for any $i, j \in \{1, \ldots, N\}$, and $k \in \{1, \ldots, K\}$.

$$w_{i,j,k} = \pi \left( \hat{w}_{i,j,\sigma^{(i)}(k)} \right), \quad w_{i,j,k} \in \{1, \ldots, n_\epsilon - 1\}. \tag{49}$$

This implies partial identifiability with respect to the labels of conditional effects $\mathcal{W}$ (Def. A.3). $\qquad \square$

## B. Implementation Details

### B.1. ELBO Derivation

Below we derive the ELBO objective for SDCI, which is expressed in Eq. (15). We start with the likelihood term $p_\psi(x_{1:T})$ and write it in terms of the joint distribution $p_\psi(x_{1:T}, s_{1:T}, \mathcal{W})$.

$$\log p_\psi(x_{1:T}) = \log \sum_{\mathcal{W}} \sum_{s_{1:T}} p_\psi(x_{1:T}, s_{1:T}, \mathcal{W}) \tag{50}$$

$$\geq \mathbb{E}_{q_\phi(\mathcal{W}, s_{1:T} | x_{1:T})} \left[ \log \frac{p_\psi(x_{1:T}, s_{1:T}, \mathcal{W})}{q_\phi(\mathcal{W}, s_{1:T} | x_{1:T})} \right] \tag{51}$$

$$\geq \mathbb{E}_{q_\phi(\mathcal{W}, s_{1:T} | x_{1:T})} \left[ \log \frac{p_\psi(x_{1:T}, s_{1:T} | \mathcal{W})}{q_\phi(s_{1:T} | x_{1:T})} \right] + \mathbb{E}_{q_{\phi_w}(\mathcal{W} | x_{1:T})} \left[ \log \frac{p_{\psi_w}(\mathcal{W})}{q_{\phi_w}(\mathcal{W} | x_{1:T})} \right] \tag{52}$$

$$\geq \mathbb{E}_{q_\phi(\mathcal{W}, s_{1:T} | x_{1:T})} \left[ \log p_\psi(x_{1:T} | s_{1:T}, \mathcal{W}) \right] - KL \left( q_{\phi_w}(\mathcal{W} | x_{1:T}) \middle\| p_{\psi_w}(\mathcal{W}) \right) \tag{53}$$

$$+ \mathbb{E}_{q_{\phi_s}(s_{1:T} | x_{1:T})} \left[ \log \frac{\prod_{t=1}^{T} p_{\psi_s}(s_t | x_{t:t-L_x}, s_{t-1:t-L_s})}{q_{\phi_s}(s_{1:T} | x_{1:T})} \right] \tag{54}$$

$$\geq \mathbb{E}_{q_\phi(\mathcal{W}, s_{1:T} | x_{1:T})} \left[ \log p_{\psi_x}(x_{1:T} | s_{1:T}, \mathcal{W}) \right] - KL \left( q_{\phi_w}(\mathcal{W} | x_{1:T}) \middle\| p_{\psi_w}(\mathcal{W}) \right) \tag{55}$$

$$- \sum_{t=1}^{T} KL \left( q_{\phi_s}(s_t | x_{1:T}) \middle\| p_{\psi_s}(s_t | x_{t:t-L_x}, s_{t-1:t-L_s}) \right) \tag{56}$$

### B.2. Exact Inference on State Variables

Exact inference on state variables based on sum-product implementations require computing the posterior for all combinations of states at each time step, thus introducing $\mathcal{O}(K^N)$ complexity. However, in the determined states case, the state variable is directly determined by observations, which implies $p(s_{t-1}^{(i)} | x_{1:t-1}) = p(s_{t-1}^{(i)} | x_{t-1}^{(i)})$. To compute the reconstruction term in Eq. (15), consider the likelihood distribution $p(x_{1:T} | \mathcal{W}) = \prod_{t=1}^{T} p(x_t | x_{1:t-1}, \mathcal{W})$, where we assume the states have been marginalised. Following our implementation, consider the likelihood term $p(x_t | x_{1:t-1}, \mathcal{W})$ and assume the message $\mathbf{h}_t^{(ij)}$ follows delta distributed variable. Expanding the marginalisation equation of the state variables results in the following:

$$p_\psi(x_t | x_{1:t-1}, \mathcal{W}) = \prod_j p_{\psi_x}(x_t^{(j)} | \mathbf{h}_t^{(1j)}, \ldots, \mathbf{h}_t^{(Nj)}, x_{t-1}^{(j)}) \prod_{i \neq j} \left( \sum_{s_i^{t-1}} p_{\psi_x}(\mathbf{h}_{ij} | x_{t-1}^{(i)}, x_{t-1}^{(j)}, \mathcal{W}, s_{t-1}^{(i)}) p_{\psi_s}(s_{t-1}^{(i)} | x_{t-1}^{(i)}) \right) \tag{57}$$

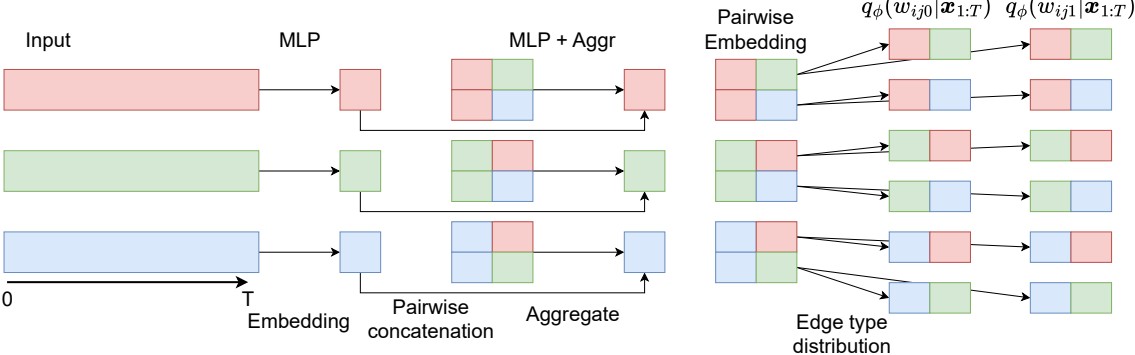

Figure 13: Illustration of the implementation of the SDCI encoder which is adapted from ACD (Löwe et al., 2022) and allow for conditioning on states. In the example, we consider 2 states.

Note that we can marginalise the states element-wise, thus reducing the exponential cost to $\mathcal{O}(NK)$. Given the above equation is equivalent to Eq. (16), we can set $q_{\phi_s} = p_{\psi_s}$ for exact inference on state variables.

For the recurrent state case, we cannot simplify the forward state posterior $p_\psi(s_{t-1}^{(i)}|\boldsymbol{x}_{1:t-1})$. To see why exact inference here results in exponential cost, compute $p_\psi(\boldsymbol{x}_t|\boldsymbol{x}_{1:t-1})$, where we can also try marginalising the states.

$$p_\psi(\boldsymbol{x}_t|\boldsymbol{x}_{1:t-1}, \mathcal{W}) = \prod_j p_{\psi_x}(\boldsymbol{x}_t^{(j)}|\mathbf{h}_t^{(1j)}, \ldots, \mathbf{h}_t^{(Nj)}, \boldsymbol{x}_{t-1}^{(j)}) \prod_{i \neq j} \left( \sum_{s_i^{t-1}} p_{\psi_x}(\mathbf{h}_{ij}|x_{t-1}^{(i)}, x_{t-1}^{(j)}, \mathcal{W}, s_{t-1}^{(i)}) p_\psi(s_{t-1}^{(i)}|\boldsymbol{x}_{1:t-1}^{(i)}) \right).$$
(58)

Where the state posterior is obtained using the forward algorithm:

$$p_\psi(s_{t-1}^{(i)}|\boldsymbol{x}_{1:t-1}) = \sum_{s_{t-2}^{(i)}} p_\psi(s_{t-1}^{(i)}, s_{t-2}^{(i)}|\boldsymbol{x}_{1:t-1}) = \sum_{s_{t-2}^{(i)}} p_{\psi_s}(s_{t-1}^{(i)}|s_{t-2}^{(i)}, \boldsymbol{x}_{t-1:t-M}) p_\psi(s_{t-2}^{(i)}|\boldsymbol{x}_{1:t-1})$$
(59)

where $p_{\psi_s}(s_{t-1}^{(i)}|s_{t-2}^{(i)}, \boldsymbol{x}_{t-1:t-M})$ is given from the state decoder, and $p_\psi(s_{t-2}^{(i)}|\boldsymbol{x}_{1:t-1})$ is computed as follows:

$$p_\psi(s_{t-2}^{(i)}|\boldsymbol{x}_{1:t-1}) = \frac{p_\psi(s_{t-2}^{(i)}, \boldsymbol{x}_{1:t-2}) p_\psi(\boldsymbol{x}_{t-1}|s_{t-2}^{(i)}, \boldsymbol{x}_{t-2})}{p_\psi(\boldsymbol{x}_{1:t-2}) p_\psi(\boldsymbol{x}_{t-1}|\boldsymbol{x}_{1:t-2})} = p_\psi(s_{t-2}^{(i)}|\boldsymbol{x}_{1:t-2}) \frac{p_\psi(\boldsymbol{x}_{t-1}|s_{t-2}^{(i)}, \boldsymbol{x}_{t-2})}{p_\psi(\boldsymbol{x}_{t-1}|\boldsymbol{x}_{1:t-2})},$$
(60)

and we find recursive terms $p_\psi(\boldsymbol{x}_{t-1}|\boldsymbol{x}_{1:t-2})$ and $p_\psi(s_{t-2}^{(i)}|\boldsymbol{x}_{1:t-2})$. Unfortunately, computing $p_\psi(\boldsymbol{x}_{t-1}|s_{t-2}^{(i)}, \boldsymbol{x}_{t-2})$ requires marginalising $(s_{t-2}^{(1)}, \ldots, s_{t-2}^{(N)})$ and involves all possible tuple combinations, thus introducing computational cost of $\mathcal{O}(K^N)$.

$$p_\psi(\boldsymbol{x}_{1:t}, \boldsymbol{s}_{t-1}|\mathcal{W}) = p(\boldsymbol{x}_t|\boldsymbol{x}_{t-1}, \boldsymbol{s}_{t-1}, \mathcal{W}) \sum_{\boldsymbol{s}_{t-2}=k} p(\boldsymbol{s}_{t-1}|\boldsymbol{s}_{t-2}=k, \boldsymbol{x}_{t-1}) p(\boldsymbol{s}_{t-2} =, \boldsymbol{x}_{1:t-1}, \mathcal{W})$$
(61)

### B.3. Encoder Architecture

Below we provide details our encoder architecture $q_\phi(\mathcal{W}, \boldsymbol{s}_{1:T}|\boldsymbol{x}_{1:T}) = q_{\phi_w}(\mathcal{W}|\boldsymbol{x}_{1:T}) q_{\phi_s}(\boldsymbol{s}_{1:T}|\boldsymbol{x}_{1:T})$.

**Fixed graph encoder.** As mentioned, we consider an amortised encoder on $\mathcal{W}$ for fixed state-dependent structures across samples: $q_{\phi_w}(\mathcal{W}|\boldsymbol{x}_{1:T}) = q_{\phi_w}(\mathcal{W})$. We modify Eq. (13) and directly parametrise the logits $\phi_{ij} \in \mathbb{R}^{K \times n_\epsilon}$ independently of $\boldsymbol{x}_{1:T}$.

**Variable graph encoder** Our architecture extends directly from ACD (Löwe et al., 2022), and we incorporate additional edge-edge layers proposed in Chen et al. (2021) only for SDCI under the recurrent state case. The logits $\phi_{ij}$ for the

distribution $q_{\phi_w}(\mathcal{W}|\boldsymbol{x}_{1:T})$ are obtained as follows. First, the model computes a latent embedding $\boldsymbol{z}_1^{(i)}$ for each node $i$ using the whole sequence:

$$\boldsymbol{z}_1^{(i)} = f_{\phi_1}(\boldsymbol{x}_{1:T}^{(i)}). \tag{62}$$

Then each embedding is updated using a graph neural network (GNN) that captures the correlations between nodes. Specifically the message passing procedure follows the two equations below:

$$\boldsymbol{z}_1^{(ij)} = f_{\phi_2}(\boldsymbol{z}_1^{(i)}, \boldsymbol{z}_1^{(j)}), \tag{63}$$

$$\boldsymbol{z}_2^{(i)} = f_{\phi_3}\Big(\sum_{i \neq j} \boldsymbol{z}_1^{(ij)}\Big). \tag{64}$$

For the recurrent state case, we incorporate edge-to-edge message passing operations proposed by Chen et al. (2021) after computing the edge embeddings in Eq. (63). The edge-to-edge message passing treats the set of edges as a sequence, and we implement the mapping using self-attention. See Chen et al. (2021) for additional details. Finally, we obtain the softmax logits $\boldsymbol{\phi}_{ij} \in \mathbb{R}^{K \times n_\epsilon}$ for every pair $i \to j$ and every state value $1 \leq k \leq K$:

$$\boldsymbol{\phi}_{ij} = f_{\phi_4}(\boldsymbol{z}_2^{(i)}, \boldsymbol{z}_2^{(j)}). \tag{65}$$

The above network architecture design is visualised in Figure 13. according to equation 13. The details of the architecture settings follows the designs in Löwe et al. (2022) and Chen et al. (2021). Each embedding step $f_{\phi_i}$, including the self-attention network in the edge-to-edge message passing, uses two-layers of 256 dimensions and ELU (Clevert et al., 2016) activations followed by a batch normalization. $f_{\phi_4}$ uses skip connections and we modify its output size to generate a pairwise embedding for each of the $K$ states. For fully-observed state case, the architecture for $q_{\phi_w}(\mathcal{W}|\boldsymbol{x}_{1:T}, \boldsymbol{s}_{1:T})$ follows a similar structure, except that for the first layer we use $\boldsymbol{z}_2^{(i)} = f_{\phi_1}(\text{concat}(\boldsymbol{x}_{1:T}^{(i)}, \hat{\boldsymbol{s}}_{1:T}^{(i)}))$, where $\hat{\boldsymbol{s}}_{1:T}^{(i)}$ is a one-hot encoded version of the states $\{s_t^{(i)}\}_{t=1}^T$.

**State encoder.** For determined states, we use the state decoder explained in the next paragraph. For recurrent states, we combine edge embeddings and the state posterior approximation from Ansari et al. (2021) to implement a GNN-RNN network. First, we generate edge embeddings with temporal components. This is equivalent to Eq. (62) where we use $\boldsymbol{x}_t^{(i)}$ to obtain $\boldsymbol{z}_{1,t}^{(i)}$. Then, we obtain edge embeddings $\boldsymbol{z}^{(ij)}{}_{1,t}$ similarly as in Eq. (63) using the same implemntation design. The temporal edge are forwarded to a 3-layer bi-directional RNN with GRU cells and 256 dimensions, followed by a 1-layer forward RNN with GRU cells and 256 dimensions. Finally, the resulting embeddings are aggregated following Eq. (65) with similar implementation designs.

**State Decoder.** The state decoder is implemented as a two-layer MLP with 256 dimensions, where the input is dependent on the requirements for each state dependency case: $\boldsymbol{x}_t^{(i)}$ in the determined case, and $(\boldsymbol{x}_t^{(i)}, \boldsymbol{x}_{t-1}^{(i)}, s_{t-1}^{(i)})$ in the recurrent case.

**Observation Decoder.** We implement the set of functions $\mathcal{F} = \{f_1, \dots, f_{n_{\epsilon-1}}\}$ using two-layer MLPs of 256 dimensions and leaky-ReLU activations with slope $0.1$. The mapping $g$ uses skip-connections and the same design and the edge-type functions above. For gene regulatory networks, we use incorporate node embeddings into the aggregator $g$ following Gong et al. (2023), resulting in increased network expressivity. The node embeddings have 16 dimensions. For NBA data, we instead incorporate the ball's features at each time step $\mathbf{b}_t \in \mathbb{R}^6$, denoting 3D position and velocity.

### B.4. Training Specifications

Our method has been implemented with Pytorch (Paszke et al., 2019), and the experiments have been performed on NVIDIA RTX 2080 Ti GPUs. Code for our experiments is available at `https://github.com/charlio23/SDCI`.

**SDCI and ACD.** All SDCI and ACD (Löwe et al., 2022) models have been trained using the following training scheme. Following Kipf et al. (2018), the models are trained using ADAM optimizer (Kingma & Ba, 2015). In all datasets, the learning rate of the edge labels encoder is $5 \cdot 10^{-4}$ for variable graphs, and $5 \cdot 10^{-3}$ for fixed graphs. The learning rate of the decoder is $5 \cdot 10^{-4}$. Learning rate decay is in use with factor of $0.5$ every 200 epochs. The duration of the training phase differs between datasets and state dependence considerations. We monitor the reconstruction error on validation data and use early-stopping. We list additional specifications which depend on each dataset.

- In springs and magnets data with determined states we train for 1000 epochs using a batch sizes of 100, 50, and 20 for $N$ being 5, 10, and 20 respectively. For recurrent states, the GNN-RNN network restricts GPU capacity, and we adapt the number of epochs and batch size to perform 200k iterations, and decaying the learning rate every 60k iterations.

- In gene regulatory network data, in the determined state setting we reduce the batch size to 10 to meet GPU capacity and train for 1000 epochs. For recurrent states, we lower the batch size to 1 due to the increased number of variables, and train for 200 epochs (200k iterations), similarly to the springs and magnet case. In all settings, we use 2 edge-types and a sparsity regularisation of 0.9 using $p_{\psi_w}$.

- In NBA data, we train for a total of 450k iterations due to the large number of samples, with a batch size of 100 and decaying the learning rate every 200k iterations. We find both SDC and ACD perform best using 2 edge-types and a sparsity regularisation of 0.8 using $p_{\psi_w}$.

The decoder is trained with teacher forcing every 10 time-steps, i.e., it receives the ground-truth as input every 10 time-steps. The variance of the decoder Gaussian distribution is $\sigma^2 = 5 \cdot 10^{-5}$. The temperature term of the edge label encoder $\tau$ is set to 0.5. The state encoder temperature $\gamma$ follows a schedule similar to Ansari et al. (2021) which prevents state collapse, i.e. the model ignoring states. We first set $\gamma = 5$, and decrease temperature every epoch by a factor of 0.8 until we have $\gamma = 0.5$.

**VRNN.**    The experiments with NBA player trajectories consider VRNN as a non-causal baseline to compare forecasting performance. Below we specify the network architecture and training scheme. To allow a fair comparison between SDCI, ACD, NRI, and VRNN, we modify the decoder defined in Chung et al. (2015) to condition the player positions on the ball features, similarly as we did for the previous models: $p_\theta(\boldsymbol{x}_t|\boldsymbol{x}_{<t}, \boldsymbol{z}_{\leq t}, \mathbf{b}_t)$, where $\mathbf{b}_t \in \mathbb{R}^6$ represents the ball features at time $t$ (3D position and velocity). The architecture of the model follows the original work: 3-layer LSTM networks with 256 dimensions and a latent space size of 128 dimensions. The encoder and decoder architectures use two-layer MLPs of 256 dimensions. The models are trained using ADAM (Kingma & Ba, 2015) for 350K iterations with a learning rate of $10^{-4}$ and batch size 32.

**NRI-MPM.**    We train our NBA player trajectory model by using the first 100 time steps of player and ball trajectories as input and predicting the next 100 time steps. Our experimental setup follows the settings outlined in (Chen et al., 2021). The NRI model is trained with both the encoder and decoder using GRU- and Attention-based architectures for edge type [2, 4]. Training is conducted over 500 epochs with a batch size of 128. The initial learning rate is set to $2.5 \times 10^{-5}$ and is decayed by a factor of 0.5 every 200 epochs. We use a temperature parameter $\tau = 0.5$, and all hidden layers across the model have a dimension of 256. During training, the model predicts M = 20 future time steps at each iteration.

**Rhino.**    For GRN and springs, we tune the hyperparameters of Rhino based on the validation RMSE error. Our experimental setup follows a similar configuration to (Gong et al., 2023), with the key difference that we use a single MLP layer with 15 hidden units for the function $f_i$. We train the model using a batch size of 128 and an initial learning rate of 0.001, optimized with Adam. The node embedding dimensions are set to 32 and 16 for each respective dataset. Additionally, we use lag = 1, auglag = 60, and apply a sparsity penalty $\lambda_s$ of 19 and 25 for the two datasets, respectively.

**N-GC.**    For Neural Granger causality (Tank et al., 2021), we use the official implementation based on MLPs. We set the default hyperparameter settings, which include large training runs with early stopping; and used MLPs with 256 hidden units.

**R-PCMCI.**    Regime-PCMCI does not support batch mode. This implies the algorithm needs to be re-trained for each sample, and we take the average summary graph across all the samples. Given available official code implementation (Saggioro et al., 2020), exploring hyperparameter settings resulted into errors from training with short sequences ($T = 50$). We use the predefined hyperparameters, and select 4 regimes to align with the number of components in the backbone state network used in GRN data.

## C. Datasets

In this section we provide detailed information about the datasets used in this work. For spring data, we generate 10000 samples of each setting for training the models. Regarding testing, we compute all the metrics using 100 samples, and we note that Rhino and Neural Granger Causality require retraining for data with variable graphs across samples.

## C.1. Springs and Magnets

When considering springs and magnets with directed connections, we follow the generation procedure described Kipf et al. (2018) with a small modification where the spring interaction between a pair of particles can change over time (depending on state variables), and the addition of magnetic interactions. This assumes a setup with $n_\epsilon = 3$, describing no-edge, and spring or magnetic effects.

In this dataset, $N$ particles are simulated inside a 2D box where they can collide elastically with its walls. Each pair of variables is connected by a spring with uniform probability. To allow for identification of causal connections (directed edges), the connection is made unidirectional. The spring interaction follows the Hooke's law, and the magnetic interaction follows a standard repulsive magnetic pole model. These interactions are captured in terms of third order moments and aggregated following the second Newton's law by summation of forces:

$$\mathbf{f}_{ij} = -\mathbf{1}_{(w_{ijk}=1)}(\mathbf{r}_i - \mathbf{r}_j) + \mathbf{1}_{(w_{ijk}=2)}\frac{1}{4\pi}\frac{(\mathbf{r}_i - \mathbf{r}_j)}{||\mathbf{r}_i - \mathbf{r}_j||^2}, \quad \ddot{\mathbf{r}}_i = \sum_{j=1}^{N}\mathbf{f}_{ij}, \quad \boldsymbol{x}_i = \{\mathbf{r}_i, \dot{\mathbf{r}}_i\}, \quad \mathbf{r}_i \in \mathbb{R}^2, k = s_t^{(i)}, \quad (66)$$

where edge-types 1 and 2 are associated to spring and magnetic effects respectively. $\mathbf{f}_{ij}$ is the unidirectional interaction from particle $j$ to particle $i$, $\mathbf{r}_i$ and $\dot{\mathbf{r}}_i$ denote the 2D position and velocity of each particle. The continuous variable $\mathbf{x}_i$ is constructed by concatenating the position and the velocity measurements. Notice that the above equation defines the evolution of the continuous variable for a single time-step. In our setting, we have that $k = s_t^{(j)}$. Thus, $\mathbf{f}_{ij}$ will change over time, contrary to Kipf et al. (2018).

To generate samples, we first generate random labels of conditional effects $\mathcal{W}$ and the initial location and velocity. The sparsity of the structure is set to 0.5, 0.7, and 0.8 when considering 5, 10, and 20 variables respectively. Then, trajectories are simulated by solving the previous differential equations using leapfrog integration. The step size used is 0.001 and the trajectories are obtained by sub-sampling each 100 steps. In our experiments, we set $T = 80$. When considering determined, we set $s_i^t = \mathbf{1}_{(\mathbf{r}_{t,0}^{(i)}>0)}$ for 2 states; and $s_i^t = \varphi(\{\mathbf{1}_{(\mathbf{r}_{t,0}^{(i)}>0)}, \mathbf{1}_{(\mathbf{r}_{t,1}^{(i)}>0)}\})$ for 4 states, where $\varphi$ injectively assigns integers to tuples. For recurrent states, we consider $K = 2$ and alternate states on wall collision.

## C.2. Gene Regulatory Networks

Gene expression data was generated with DynGen (Cannoodt et al., 2021). The simulation engine includes different types of backbone state networks, namely bifurcation, cyclic, or single line networks. We choose a bifurcating network with 4 state modules and 2 bifurcations; with a total of 49 genes, including 16 target genes and no housekeeping genes. Target genes are characterised by not taking part of the activation/deactivation regimes and only receive interactions from genes associated to states. A synchronised experiment was simulated with 50 time steps and for 1000 cells. Real sequencing experiments cannot measure the same cell over time and simulation engines typically sample different populations of cells at every time step. However, facilitating causal discovery require that the gene expression of the same cell is measured over time. To obtain this, the model's simulated mRNA counts were used directly. As mentioned, we used meta-data from the backbone state transitions as auxiliary information to SDCI in the observed case.

## C.3. NBA Data

The NBA dataset (Linou, 2016) consists of recordings from 632 NBA games played during Winter 2015-2016. Each game is composed by approximately 400 to 600 events, which represents sequences of plays. In each trajectory, we find information about the ball location and 5 players of the 2 different teams (10 in total). The coordinates of the ball and players are represented in 3D and the length of each sequence can vary from 100 to 600. In our experiments, we consider a sequence up to 200 time-steps ($T = 100$ for reconstruction and the rest 100 steps for prediction), which gives us a total training dataset of 150K samples and a test set of 6380 samples. Data inspection shows that the court size is $100 \times 50$, and we use this information to standardise the data. For experiments with SDCI with observed states, we design some ground-truth states which depend on different locations of the court. We set $K = 4$ and our choice is shown in figure 14.

---

[2]Data extracted from the following code repository `https://github.com/linouk23/NBA-Player-Movements` – last accessed 2022-09-28.

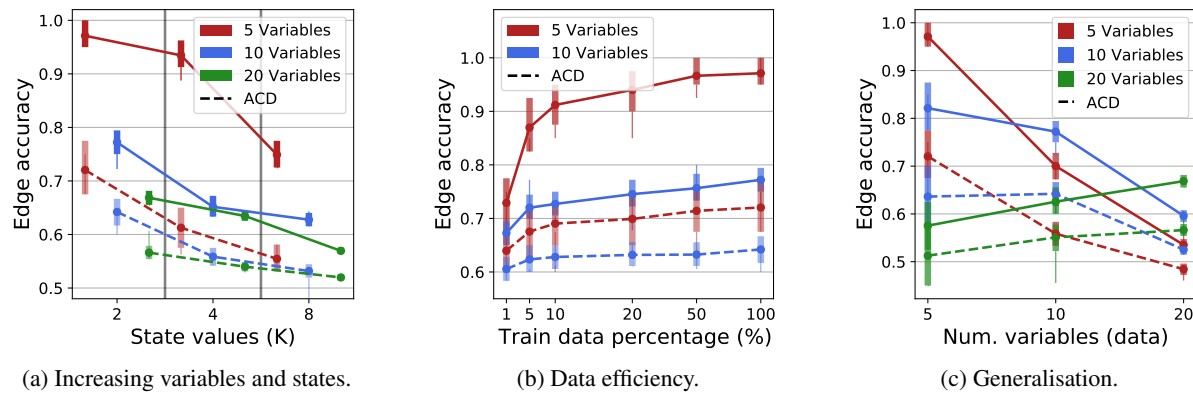

(a) Increasing variables and states.      (b) Data efficiency.      (c) Generalisation.

Figure 15: Spring data with observed states for (a) increasing variables and state values, (b) data efficiency, and (c) generalisation, where the x-axis indicates the test data and the legend indicates the model used. SDCI results are shown with solid lines.

## D. Comments on Evaluation Metrics

### D.1. Summary Graph Estimation

For multi-dimensional data such as springs, we estimate the summary graph of Rhino and N-GC by grouping elements together, and for each element we include an edge if any of its dimensions has interactions with other elements.

### D.2. $F_1$ Scores

To clarify, the evaluation of the summary graph considers the micro averaged $F_1$ score, to account for increased sparsity on increasing variables. For edge labels $\mathcal{W}$ and states $s_{1:T}$, we use macro averaged $F_1$ score, as no restrictions on proportion of labels are made.

### D.3. Computing the Summary Graphs

Notice that SDCI can extract the conditional summary graph (CSG) whereas the baselines we compare with only consider the summary graph (SG). Consequently, the only immediate way to compare the performance in capturing the causal structure among the methods we consider is to evaluate the latter. From the definition of summary graph, we deduce that one can estimate it by taking the union of the graphs in the CSG. This is used to compute the summary graphs of both SDCI and the ground truth structure of the generative process.

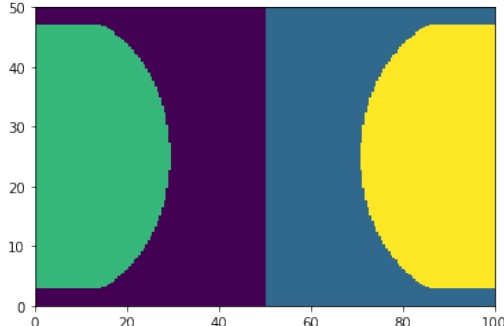

Figure 14: Hand-crafted ground-truth state map on NBA data which is forwarded to SDCI with observed states. Colours indicate different states.

## E. Additional Results

In this section we report additional experiments and qualitative visualisations, which can be helpful to complement the main results from Section 6 in the main text..

### E.1. Springs and Magnets

**Observed** In this experiment the states are known and independent from the observations. For the ground truth dynamics, the state transitions incrementally into the next one every 10 time-steps. We report accuracy with respect to the edge lables $\mathcal{W}$, and consider 2 edge-types. Figure 15a shows results with increasing variables and increasing states, where we compare with ACD (dashed lines) as a stationary baseline. Although performance drops as $K$ increases, SDCI is able to maintain reasonable accuracy in edge-type identification in comparison to the stationary baseline. With increasing number of variables both approaches see accuracy drops; our hypothesis is that this can be addressed by increasing model capacity. The next test

considers data efficiency of SDCI with results reported in Figure 15b. We see that both SDCI and ACD are data efficient in this scenario, where training on 10% of the data returns good performance already. Finally, we also report in Figure 15c on generalisation to unseen data with different number of variables. Here both methods generalise better to settings where the number of variables is similar to the ones they were trained.

**Determined states** We provide visualisations for the determined states case in Figure 17, where we show the predictions of both SDCI and ACD as well as the corresponding conditional summary graphs and summary graphs extracted by both methods respectively. We observe SDCI produces accurate causal graph estimates. Regarding time series forecasting, our method is able make reasonable predictions. Notice that to train the models, we use teacher forcing every 10 time-steps, which means that the learned models are less suited for long-term dynamics modelling. However, one can expect to obtain more accurate predictions by progressively reducing the teacher forcing frequency during training. Considering ACD, despite being restricted by assuming stationary time series, it still infers graph structures that allow the model to produce adequate forecasts.

### E.2. NBA Data

To further analyse the representations learned by SDCI, we visualise additional examples in Figure 16. The extracted graphs exhibit similar patterns to those presented in the main text. In the first example, the blue team is on offense, and we observe a higher presence of edges from the defending team. A similar pattern can be seen in the second example, where the red team takes the offensive role instead. Overall, SDCI assigns regimes with distinct sparsity patterns, effectively adapting to the nonstationary nature of the data.

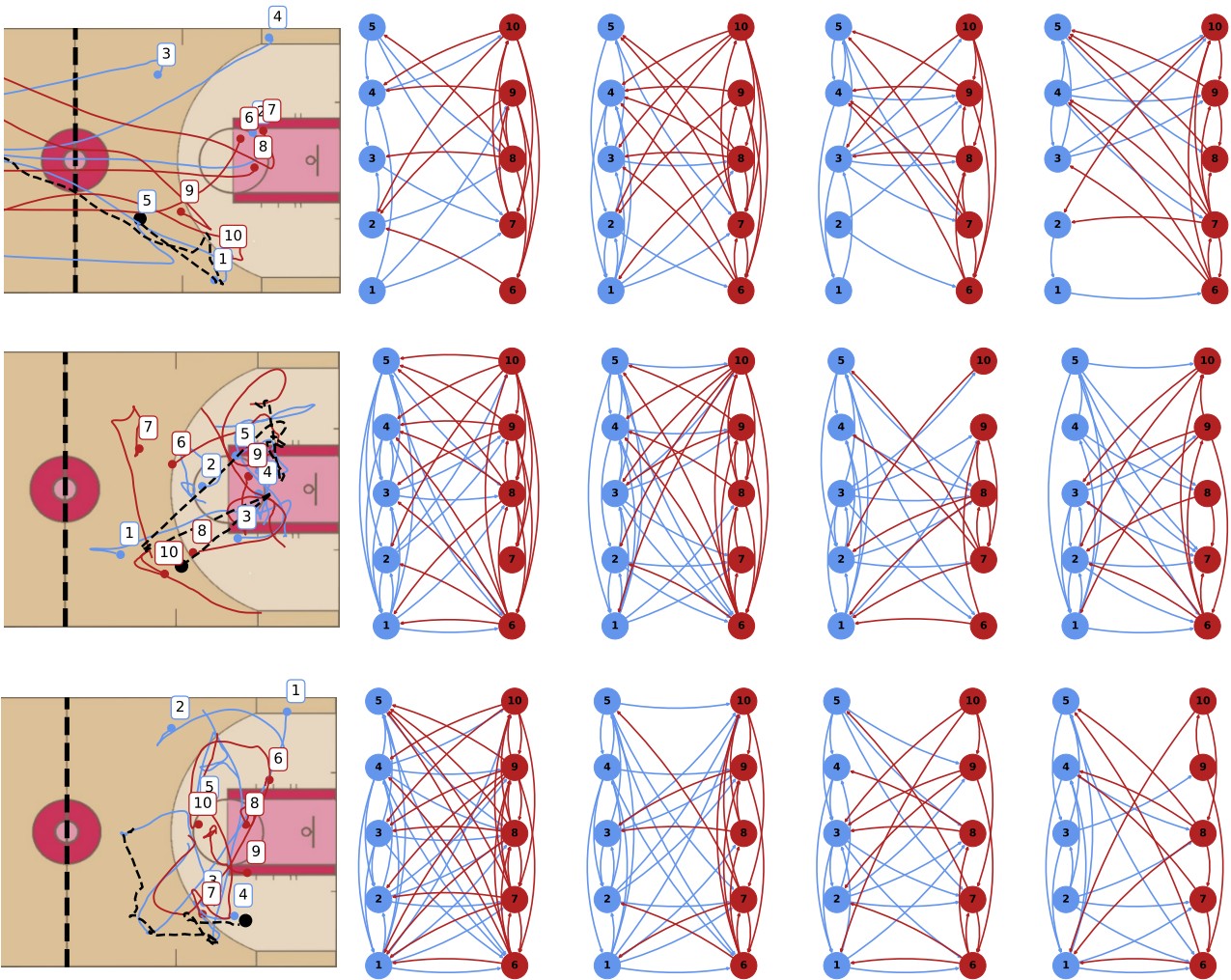

Figure 16: Examples of extracted conditional summary graphs from SDCI for $K = 4$, with states determined by the position and velocity of the players. Each row describes a different sample.

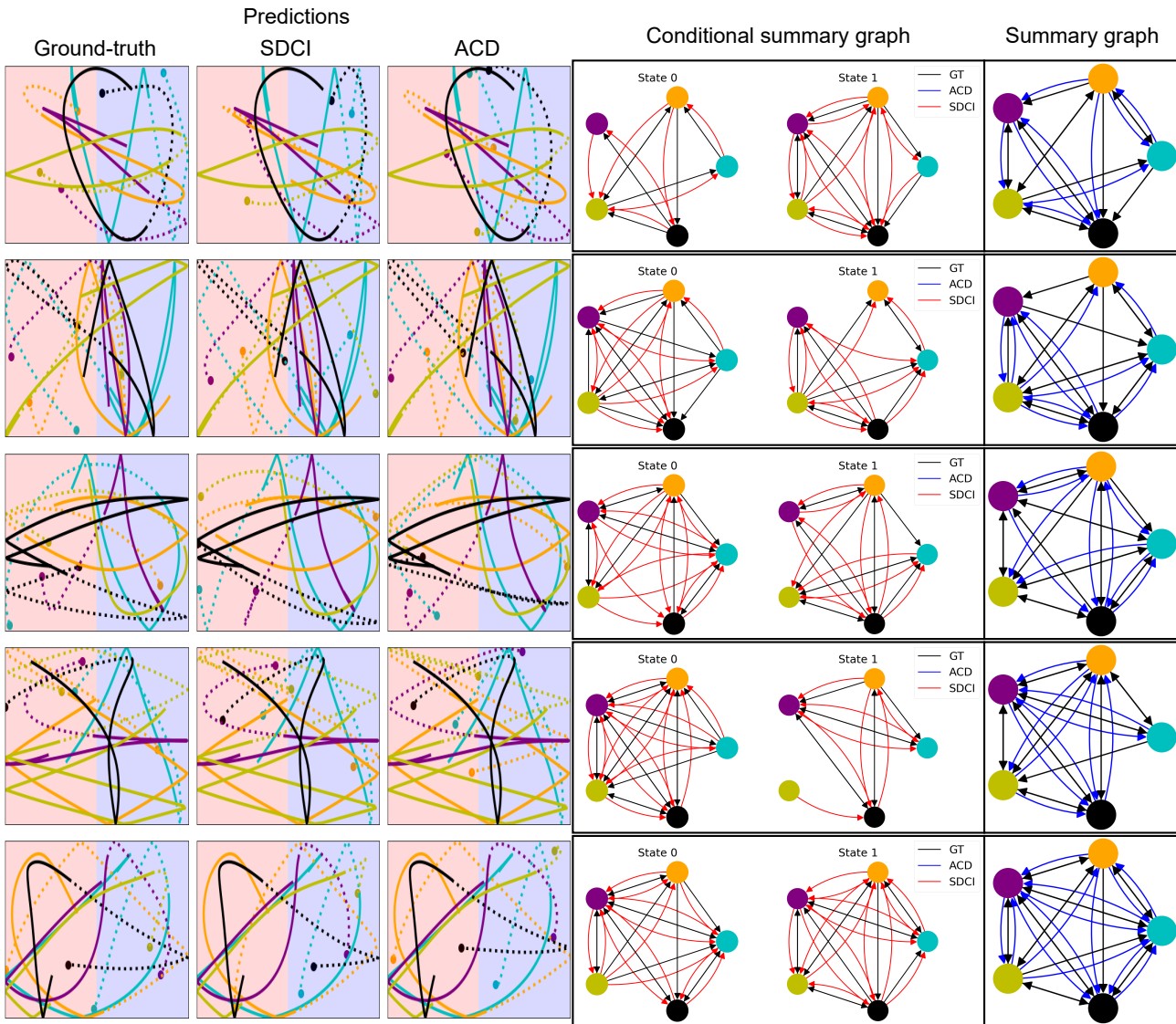

Figure 17: time series forecasting (left, dotted lines) of SDCI and ACD for 50 time-steps along with the ground-truth. We use solid lines to denote the input to the models and the background color represents the state value. We show the associated conditional summary graph (center) and summary graph (right) of SCDI (red) and ACD (blue) respectively along with the ground-truth (GT, black) for each sample. Each row represents a different sample.

