# OpenReview forum: "Causal Discovery from Conditionally Stationary Time Series"
_ICML.cc/2025/Conference — ICML 2025 poster_

### Official Review · Reviewer_5FGR · 2025-03-11

**Overall Recommendation:** 3

**Summary:**

This paper extends the existing work in time series causal discovery from stationary data to conditionally stationary time series data, which is stationary if conditioned on the latent states. The authors propose a conditional summary graph to represent the causal structure and give the identifiability results. Empirical experiments on semi-synthetic and real-world data show the superior performance over the baselines.

## update after rebuttal
I will keep my score after reading the response and the other reviews.

**Claims And Evidence:**

The claims made in the submission (e.g. establishing the identifiability for the conditional summary graph and related structural properties) are supported by the clear definition (Def. 4.1), reasonable assumptions, and rigorous theoretical results (Thm. 4.3). The experimental results, conducted under the authors’ specified settings, provide convincing empirical validation. However, providing more real-world examples for the three classified scenarios of states would improve the paper’s practical relevance.

**Essential References Not Discussed:**

I did not identify any missing essential references, though further discussion on related work in causal identifiability for time series data could be beneficial.

**Experimental Designs Or Analyses:**

The experimental design is well-structured and appropriate, leveraging both semi-synthetic and real-world data. Careful modifications and adjustments have been applied to the datasets to ensure smooth comparisons. However, the evaluation criteria could be clarified further. For example, in Fig. 6, does the fixed graph correspond to the summary graph for the full-time graph, while the variable graph represents the conditional summary graph? Explicit clarification of these evaluation targets would enhance the interpretability of the results.

**Methods And Evaluation Criteria:**

Overall, the proposed State-Dependent Causal Inference (SDCI) framework makes sense for the problem of causal discovery in conditionally stationary time-series data. The probabilistic formulation, graph-based modeling, and variational inference provide a compelling approach. However, the clarity of the notations and the derivations of the formulations could be further improved.

**Other Comments Or Suggestions:**

Some typos:
1.	Line 146\~147, right side, “the parent of $x^{(2)}\_2$ are …” lost a “{“.
2.	Line 217, left side,  “we can guide $\psi_\omega$ though” -> through.
3.	Line 738 (a2.2), “, the following set has zero measure:”, where is the following set?
4.	Line 1022~1023, what’s behind “=”?
5.	The citation format is inconsistent in the Related Work part.
I found the notations not that clear, e.g., an explicit footnote about $j\in C_k^{(i)} \iff x_{t-1}^{(i)}\in PA_{s_{t-1}}^{(j)}(t-1), s_{t-1}=k$ and the reason why PA should take t-1 as input but $C$ should not, might be rather helpful.

**Other Strengths And Weaknesses:**

Strengths:
1.	This paper is original for its proposed setting (stationary time-series when conditioned on states). Based on the identifiability of Markov switching models, the identifiability of this model is also guaranteed.
2.	This work is significant since it extends the stationary time-series to a broader setting.

Weaknesses:
1.	There is room for the clarity/simplification of the notations. For example,  it is hard to differentiate a real probability density from an estimated one. Especially in Section 3.3, $p$, $p_{\psi}$, and $q_{\phi}$ confused me.
2.	The work seems like a compact version of the regime-dependent graph, which also considers the time-series data in a conditional stationary setting. The identifiability results are impressive, but I am concerned with the contributions.

**Questions For Authors:**

1.	Is the proof of Thm. A.6 originally derived by the authors, or is it adapted from existing work?
2.	Why do you propose labels of conditional effects? It seems just to work as a compact representation of the existence of edges. Could you please give a more intuitive explanation for $f_{e_i}$ in Eq. (8)?
3.	What’s the difference between two $f_{s_{t-1}}^{(j)}$s in Eq. (5) and Eq. (8)? The authors claim Eq. (8) reduces the exponential complexity of Eq. (5), but it’s bearly clear to me.
4.	How did you learn the $K$ in your implementation?

**Relation To Broader Scientific Literature:**

This work is related to causal discovery for nonstationary time-series and multi-domain data (Yao, 2022; Varambally, 2024; Song, 2024; Huang, 2015, 2019, 2020; Gong, 2022; Zhang, 2017; Ghassami, 2018). The conditional summary graph proposed by the authors provides a more compact representation than the full-time and regime-dependent graphs. The identifiability results over hidden latent states are novel.

References:
1. Yao, W., Sun, Y., Ho, A., Sun, C., and Zhang, K. Learning temporally causal latent processes from general temporal data. In International Conference on Learning Representations, 2022.
2. Varambally, S., Ma, Y., and Yu, R. Discovering mixtures of structural causal models from time series data. Proceedings of the 41st International Conference on Machine Learning, volume 235 of Proceedings of Machine Learning Research, pp. 49171–49202. PMLR.
3. Huang, B., Zhang, K., and Scholkopf, B. Identification of time-dependent causal model: A gaussian process treatment. In Twenty-Fourth international joint conference on artificial intelligence, 2015.
4. Huang, B., Zhang, K., Gong, M., and Glymour, C. Causal discovery and forecasting in nonstationary environments with state-space models. In International Conference on Machine Learning, pp. 2901–2910. PMLR, 2019.
5. Huang, B., Zhang, K., Zhang, J., Ramsey, J. D., SanchezRomero, R., Glymour, C., and Scholkopf, B. Causal discovery from heterogeneous/nonstationary data. J. Mach. Learn. Res., 21(89):1–53, 2020.
6. Gong, W., Jennings, J., Zhang, C., and Pawlowski, N. Rhino: Deep causal temporal relationship learning with history-dependent noise. In NeurIPS 2022 Workshop on Causality for Real-world Impact, 2022.
7. Zhang, K., Huang, B., Zhang, J., Glymour, C., and Scholkopf, B. Causal discovery from nonstationary/heterogeneous data: Skeleton estimation and orientation determination. In IJCAI: Proceedings of the Conference, volume 2017, pp. 1347. NIH Public Access, 2017.
8. Ghassami, A., Kiyavash, N., Huang, B., and Zhang, K. Multi-domain causal structure learning in linear systems. Advances in neural information processing systems, 31, 2018.

**Theoretical Claims:**

Overall, the proofs provided for the Thm. A.6, Thm. A.7, and Corollary A.8 seem correct. The reasoning effectively utilizes key assumptions to establish partial identifiability up to permutation equivalence. Some minor refinements could further enhance accessibility and readability.

1.	The theorem appears to rely on several existing results. Providing clearer references or explanations in the appendix regarding the specific works used would enhance clarity and attribution.

2.	Additionally, for the identifiability of finite Gaussian mixture, does it need any regularity assumptions? Please clarify these assumptions if you are going to use the results.

---

> ### Author Rebuttal · Authors · 2025-03-28
>
> We sincerely appreciate the reviewer’s feedback. Below, we address key concerns.
> >  There is room for the clarity/simplification of the notations.
>
> We appreciate this point and will revise the manuscript for clarity.
> - Subscripts $\psi$ and $\phi$ denote parameters of the generative model and variational approximation, respectively. The generative model should always be indexed by $p_{\psi}$
> - The outgoing structure $C_k^{i}$ contains variable indices, $1, \dots, N$, while $PA_{s_{t-1}}(t-1)$ refers to function arguments (e.g. $x_{t-1}^{(i)}$). For time series, we consider the time index when computing densities at each time step $t$, and this is why we use $t-1$ as an argument to $PA$.
>
> We will make sure the above is revised and address all noted typos.
> > but I am concerned with the contributions.
>
> Our theoretical results encompass three levels of generality.
> - Markov Switching Models (m1-m3): Thm. A.6
> - Conditionally stationary time series (m1-m4): Thm A.7
> - SDCI (m1-m5): Cor. A.8.
>
> Our paper explicitly studies the applications of SDCI to address nonstationarity. However, our scope in terms of theoretical results is **broader**, and our results from Thms. A.6 and A.7 could be of independent interest. Identifiability in regime-dependent time series is a very challenging problem with only very recent contributions [1], and importantly, our work makes no assumptions on state dependence.
>
> Regarding empirical contributions, Gene Regulatory Network (GRN) inference methods are typically validated with semi-synthetic data that mimics the underlying biological system, as validation through real-world cell systems is very expensive. SDCI design agrees with GRN literature and its ability to model gene activation/deactivation could be impactful in this domain.
>
> [1] Balsells-Rodas et al., "On the identifiability of switching dynamical systems." ICML 2024.
>
> **Theoretical Claims and Q1**
>
> All proofs are original. Theorem A.6, leverages Yakowitz & Spragins (1968) [2], also used in [1]. We note **key differences**:
> - We allow dependencies from $x_{1:t}$ to $s_t$, strictly prohibited in [1].
> - Our weaker assumptions require a **novel proof strategy** resolving permutation equivalence issues from Yakowitz & Spragins (1968) result.
> - Our strategy addresses identifiability from **conditional distributions** $p(x_t| x_{1:t-1})$, while [1] considers the **joint distribution** $p(x_{1:T})$.
>
> Identifiability in finite mixtures relies on function linear independence [2]. For Gaussian families, this holds if means or covariances are distinct. We ensure this by assumption (a1). We will clarify the above in the appendix, and plan to include a discussion comparing other works in the main document (see Reviewer Fbj4).
>
> [2] Yakowitz and Spragins. "On the identifiability of finite mixtures." AMS 1968.
>
>
>
> **Reviewer’s questions:**
>
> Q2: Conditional effect labels improve scalability in conditionally stationary time series. SDCI, inspired by interactive systems (e.g. NRI, ACD), learns diverse graphs from limited types of interaction types $n_{\epsilon}$. Example: in sports data, different samples may have distinct graphs, but all interactions could be categorized as "no interaction," "aggressive," or "defensive."
>
> Q3:
> - Eq. (5): $f_{s_{t-1}}^{(j)}$ in is indexed by $s_{t-1}$ leading to $K^N$ functions definitions, which is prohibitively expensive.
> - Eq. (8): We use GNNs with pairwise interactions and an aggregator function, reducing complexity, and we prove identifiability of the labels under this setting.
> - SDCI considers (m1-m5), but we also prove identifiability considering weaker cases (m1-m4), allowing alternative solutions.
>
> Q4:
> - We require no knowledge of $K$, unlike most mixture model results on time series.
> - This allows finding $K$ via model selection (assuming convergence to MLE), which is theoretically impossible if $K$ requires to be known.
> - In practice, standard techniques such as the “elbow method” (used in K-means) can be applied. See the figure below for model selection with true $K=2$ in the springs and magnet data, where reconstruction MSE plateaus at $K=2$.
>
> https://postimg.cc/k2rFZkBp
>
> We plan to expand on selecting $K$ in the Experiments section, showcasing also $K=4$ in the springs and magnets data.
>
> **Actions**
>
> To improve our manuscript (see other Rebuttals for details), we aim to:
> - Clarify notation and revise typos.
> - Provide intuitions from [2] in our Proof for Thm. A.6.
> - Contrast our method assumptions with prior works, emphasizing our broader theoretical scope.
> - Expand the discussion on assumptions and implications to real-world data.
> - Provide details for $K$ selection, and clarify for NBA data.
> - Clarify scalability and include runtime analysis in Appendix.
> - Include figures for increasing data sizes on fixed graph data in Appendix.
> - Modify Figure 5b to include additional variables.
>
> We sincerely appreciate the reviewer’s insights and believe these revisions will significantly improve the paper.

---

> > ### Comment · Reviewer_5FGR · 2025-04-05
> >
> > Thank you very much for the detailed response and clarification, especially regarding the distinction from previous works and how to select $K$. As I mentioned earlier, this is a complete and well-structured paper, though the extent of its contribution is not game-changing. For this reason, I am leaning toward acceptance, and I believe my score accurately reflects my evaluation.

---

### Official Review · Reviewer_Fbj4 · 2025-03-14

**Overall Recommendation:** 2

**Summary:**

This paper introduces State-Dependent Causal Inference (SDCI), an approach for causal discovery building on VAE-based approaches in nonstationary time series characterized. SDCI leverages a "conditional summary graph" to compactly represent state-dependent causal structures and establishes identifiability guarantees under specific state-dependency assumptions.

**Claims And Evidence:**

Yes, the claims are generally supported by experiment, though there are some gaps. In particular, the paper claims that it relaxes the assumptions of causal discovery for nonstationary time series, but it lacks a clear discussion on it.

**Essential References Not Discussed:**

Most enssential references are discussed.

**Experimental Designs Or Analyses:**

Yes, the experimental designs and analyses are partial sound. Howeve, experiments are somewhat weak, e.g., the number of variables (N=3, 5) is too small in the recurrent states and varying structures experiment.

**Methods And Evaluation Criteria:**

Yes, the proposed methods and evaluation criteria are well-suited for the problem of causal discovery from nonstationary time series.

**Other Comments Or Suggestions:**

N/A

**Other Strengths And Weaknesses:**

Strengths:

- It developed a novel VAE and GNN-based framework to build the proposed method.

Weaknesses:
- The main issue is that the setting of introducing discrete latent variables to modeling nonstationary time series is not new [1]. This work claims that it relaxing assumption under such setting, but does not provide a clear discussion on how it relax the assumption. It is unclear how the proposed method is different from existing methods under the same setting.
- Although some theoretical results are developed, it is still unclear how to ensure the assumptions hold?
- Moreover, does the proposed algorithm satisfy all the requreied assumptions since there are some assumption on the function.

[1] Hälvä, Hermanni, and Aapo Hyvarinen. "Hidden markov nonlinear ica: Unsupervised learning from nonstationary time series." Conference on Uncertainty in Artificial Intelligence. PMLR, 2020.

**Questions For Authors:**

See the weaknesses above.

**Relation To Broader Scientific Literature:**

This paper related to the causal discovery literature.

**Theoretical Claims:**

The theoretical claims in this paper seems correct. However, the proposed results seem not much related to the proposed algorithm.

---

> ### Author Rebuttal · Authors · 2025-03-28
>
> We sincerely appreciate the reviewer's thoughtful comments. Below, we address the key concerns raised in the review.
>
> > It is unclear how the proposed method is different from existing methods...
>
> While prior works [1-3] also consider discrete latent variables for modeling nonstationary time series, our method has **three key advantages:**
>
> - **No assumption of known number of states:**
>   - Most previous works, including HMM-ICA [1] and CtrlNS [2], assume the **number of states is known**.
>   - Our work leverages Yakowitz & Spragins (1968) [4], which enables identifiability **without** knowing the number of states. This allows for model selection to determine $K$, an aspect theoretically impossible in prior approaches.
> - **No assumptions on state dependency:**
>   - Prior works often assume that **state transitions independent of observations** (e.g., HMM-ICA [1], [3]).
>   - Recently, NCRTL [2] models **direct observation-to-state dependencies**, but requires **stronger assumptions** (e.g., mechanism sparsity, known $K$).
>   - Our proof technique requires no assumptions in terms of state dependencies, allowing feedback from observations.
> - **Regime-dependent identifiability**
>   - While we provide identifiability for SDCI in the main text (m1-m5), our results provide a **broader scope** with general identifiability results for regime-dependent time series (m1-m3) in Theorem A.6.
>   - In Thm. A.6 we present a **novel proof technique** leveraging Yakowitz and Spragins (1968) [4]. This result can be of independent interest.
>   - Strategies based on [1] **cannot solve this problem**, as their proof strategy prohibits explicit autoregressive dependencies from $x_{t-1}$ to $x_t$.
>
> We will explicitly incorporate this discussion in the revised manuscript.
>
> [1] Hälvä, Hermanni, and Aapo Hyvarinen. "Hidden markov nonlinear ica: Unsupervised learning from nonstationary time series." UAI 2020.
>
> [2] Song, Xiangchen, et al. "Causal temporal representation learning with nonstationary sparse transition." NeurIPS 2024.
>
> [3] Balsells-Rodas, Carles, Yixin Wang, and Yingzhen Li. "On the identifiability of switching dynamical systems." ICML 2024.
>
> [4] Yakowitz, Sidney J., and John D. Spragins. "On the identifiability of finite mixtures." AMS 1968.
>
> > it is still unclear how to ensure the assumptions hold?
>
> Aligning data with assumptions is a key challenge in causal discovery and ICA. Our assumptions are as follows:
> - (m1-m5): Our model design is inspired by interactive systems, and physical models (e.g. sum of forces).
> - (a1) Unique Causal Structures Across States: Each variable must have distinct causal interactions per state.
> - (a2) Functional Faithfulness: Discussed in lines 248–253. It can be achieved by using smooth functions (e.g. analytic functions), which are commonly assumed in EEG or Gene expression data.
>
> For semi-synthetic datasets, the assumptions are simple to verify due to access to the groundtruth dynamics. In real-world data, expert knowledge is required. Assumption violations can break identifiability, which would invalidate interpretability. We discuss these issues in Section 6.3 and will expand on them in revision.
>
> > does the proposed algorithm satisfy all the required assumptions
>
> Yes, SDCI is designed to incorporate these assumptions:
> - (m1-m5) & (a2.1): Enforced via a GNN architecture and the no-edge effect (i.e. $w_{ijk}=0 \to f_0:= 0$).
> - (a1): Date-dependent, independent of algorithm design.
> - (a2.2): Ensured via analytic activation functions (e.g., Softplus, Cosine) in the edge-type mechanisms.
>
> We will enhance the discussion on how SDCI aligns with these assumptions in the final version.
>
> > The number of variables (N=3, 5) is too small
>
> See the figure below for an additional column with $N=10$ in Figure 5b. Increasing $N$ in time series causal discovery is challenging due to the temporal component.  In our setting, the total data dimensionality is $N \cdot d$ ($N\cdot 4$ in springs and magnets), but complexity also increases with $K$ and $T$.
>
> https://postimg.cc/mzznGYhz
>
> **Actions**
>
>
> To improve our manuscript (see other Rebuttals for details), we aim to:
> - Clarify notation and revise typos.
> - Provide intuitions from [2] in our Proof for Theorem A.6.
> - Contrast our method assumptions with prior works, emphasizing our broader theoretical scope.
> - Expand the discussion on assumptions and implications to real-world data.
> - Provide details for $K$ selection, and clarify NBA data.
> - Clarify scalability and include runtime analysis in Appendix.
> - Include figures for increasing data sizes on fixed graph data in Appendix.
> - Modify Figure 5b to include additional variables.
>
> We believe these revisions will strengthen the paper and appreciate the reviewer’s constructive feedback.

---

### Official Review · Reviewer_qXnG · 2025-03-17

**Overall Recommendation:** 4

**Summary:**

This paper presents a new framework for solving the causal discovery problem in non-stationary sequences. This paper makes certain assumptions under the non-stationary condition, and then proposes the new method Conditional summary graph for representing causality, and the method for conducting causal discovery state-Dependent Causal Inference. The paper provides identifiability analysis for the key steps of its approach, validating SDCI methods on semisynthetic data based on physical and biological systems and real-world datasets.

**Claims And Evidence:**

The paper mainly focuses on the causal discovery problems under new and more lenient conditions. The feasibility of the proposed method is tested through theoretical identifiability analysis and experimental testing.

**Essential References Not Discussed:**

As far as I known, it's no problem.

**Experimental Designs Or Analyses:**

The experimental design of this paper revolves around the causal discovery of conditional stationary time series, and mainly verifies the proposed SDCI framework from the simulation system (spring-magnet), gene regulation network (GRN) and real data (NBA). Covering the simulation, synthesis and real data scenarios, the physical mechanism is verified by the spring system, and the applicability of the biological network is verified by GRN, reflecting the universality of the method. The comparison methods include traditional methods (R-PCMCI, Neural Granger Causality) and leading hidden variable causal models (ACD, Rhino), covering different methodological systems. However, NBA data are only mentioned in context, lacking detailed experimental design and outcome analysis, which may weaken the credibility of the method in real scenarios. In addition, the paper mentioned that the consistency (consistency) of SDCI is still an open problem, but the targeted validation is not designed in the experiment (such as the convergence test under different data quantities), and the theoretical advantages are not fully translated into empirical results.

**Methods And Evaluation Criteria:**

The conditions assumed in this paper are mainly that the dependence of the state dependence can be decomposed into the superposition of the dependence of various features. This assumption can indeed be satisfied by prior knowledge in certain realistic scenarios. Therefore, the proposed method expands the application scope of causal discovery in the time series, which has significance.

**Other Comments Or Suggestions:**

No

**Other Strengths And Weaknesses:**

The main advantage of this paper is the method of causal discovery from conditional stable time series, which provides new ideas for solving the limitations of traditional causal inference in handling non-stable data, especially in handling causal relationships in complex systems or dynamic environment. The method is innovative in theoretical framework or algorithm design to capture more hidden causal patterns. However, the disadvantage of the paper is the strong dependence on the "conditional stability" assumption, which, if the actual data cannot meet the premise, may lead to model failure. Moreover, the computational complexity or scalability of experimental validation is not fully demonstrated.

**Questions For Authors:**

Why is it that the work of this paper does not need to know the number of K mentioned in Remark 4.2, but in experiment 3, the movement trajectory needs to do two groups: K=2 and K=4, and the results of the two groups are different?

**Relation To Broader Scientific Literature:**

All causal discovery papers require assuming certain conditions, which are often too strict for realistic scenarios to make the method inapplicable, and so many studies are devoted to relaxing these conditions. Starting from this goal, this paper studies the causal discovery problem under new and more relaxed conditions.

**Theoretical Claims:**

No

---

> ### Author Rebuttal · Authors · 2025-03-28
>
> We sincerely appreciate the reviewer’s time and thoughtful feedback. We are glad they find our framework innovative and acknowledge its significance in handling nonstationary time series. Below, we address the key concerns raised in the review.
> > ... if the actual data cannot meet the premise, may lead to model failure.
>
> SDCI is explicitly designed for conditionally stationary time series, with applications in interactive systems (e.g., sports) and Gene Regulatory Networks (GRNs). However, our theoretical results are **broader**, covering:
> - Markov Switching Models (m1-m3): Theorem A.6.
> - Conditionally stationary time series (m1-m4): Theorem A.7.
> - SDCI (m1-m5): Corollary A.8.
>
> We develop theoretical results starting from a very general case to our specific implementation, SDCI. These results, A.6 and A.7, provide a **general foundation** for regime-dependent causal discovery.
>
> We would like to note that identifiability in regime-dependent time series is a **very challenging** problem with only recent contributions [1]. Importantly, our results make no assumptions on state dependencies, and introduce a **novel proof technique** based on classic finite mixture model theory (Thm. A.6). We will incorporate a discussion emphasizing our theoretical scope in contrast with related work (see Reviewer Fbj4 for additional details).
>
> [1] Balsells-Rodas, et al. "On the identifiability of switching dynamical systems." ICML 2024.
>
>
> > … scalability of experimental validation is not fully demonstrated.
>
> We acknowledge this concern and will include additional runtime analysis in the appendix to illustrate SDCI’s linear scaling with respect to $K$. We will also compare state inference times across “determined” and “recurrent” scenarios.
>
>
> > NBA data are only mentioned in context, lacking detailed experimental design and outcome analysis
>
> We provide details on training in Appendix B.4 (lines 1097-1099) and dataset preprocessing in Appendix C.3. These details were deferred to the appendix to focus on the interpretability of learned structures.
>
> > … this paper does not need to know the number of K mentioned in Remark 4.2, but in experiment 3, the movement trajectory needs to do two groups …
>
> Our identifiability results do not require knowledge of $K$, which implies we can determine $K$ via model selection (assuming convergence to MLE). This is a significant contribution, as other related works require knowledge of $K$ (with exception of [1]) and cannot rely on this idea.
>
> However, selecting $K$ in practice is a well-known challenge even in non-temporal settings. We can use the “elbow method” (as in K-means), or similar heuristics. Below we show results for synthetic springs and magnets with $K=2$ where reconstruction MSE plateaus for $K=2$.
>
> https://postimg.cc/k2rFZkBp
>
> Our discussion regarding results on $K=2$ and $K=4$ aimed to show that both values provide meaningful interpretations. Figure 9 (forecasting results) shows similar performance for $K=2$ and $K=4$, suggesting $K=4$ overfits the data in this scenario.
>
> We acknowledge selecting $K$ is an important challenge, and will
> - include a discussion on how to select $K$ in practice, showing figures similar to the above, including a setting with $K=4$.
> - discuss how this applies to NBA data.
>
> > the targeted validation is not designed in the experiment (such as the convergence test under different data quantities) …
>
> Figure 4 shows that SDCI achieves 100% $F_1$ score when the graphs are fixed, demonstrating consistency empirically. For samples with different graphs, the challenge relies on the approximate posterior design $q(W|x_{1:T})$. The identifiability results support structure learning from purely unsupervised data, which for this setup is very challenging.
>
> We provide dataset size variation results for observed states in Appendix E.1, Figure 14. Below, we show similar results for fixed graphs on "Recurrent states" with $N=5$ and $K=2$, which will be included in the appendix:
>
> | $\|\mathcal{D}\|$              | $\mathcal{W}$ $F_1$ score | State $F_1$ score |
> | ---------------- | :---------: | :-------: |
> | 10        |   0.65   | 0.524 |
> | 100           |   0.875   | 0.543 |
> | 500    |  1.00   | 0.857 |
> | 1000 |  1.00   | 0.884 |
> | 5000 |  1.00   | 0.891 |
>
> **Actions**
>
> To improve our manuscript (see other Rebuttals for details), we aim to:
> - Clarify notation and revise typos.
> - Provide intuitions from [2] in our Proof for Theorem A.6.
> - Contrast our method assumptions with prior works, emphasizing our broader theoretical scope.
> - Expand the discussion on assumptions and implications to real-world data.
> - Provide details for $K$ selection, and clarify NBA data.
> - Clarify scalability and include runtime analysis in Appendix.
> - Include figures for increasing data sizes on fixed graph data in Appendix.
> - Modify Figure 5b to include additional variables.
>
> We appreciate the reviewer’s constructive feedback and believe these revisions will further improve the paper.

---

### Decision · Program_Chairs · 2025-05-01

**Decision:**

Accept (poster)

**Comment:**

**Summary**: The paper shows identifiability results and GNN based VAE architecture that learns multiple causal graphs one for each latent state conditioned on which the observed time series is stationar, i.e. fixed Granger causal functional relationships. Authors prove identifiability results for Markov Switching models which are mixtures of stationary time series (with sparse Granger causal relations between previous discrete time and the current time) and many other generalizations even when the unobserved latent variable support is not known.

Authors propose a GNN based VAE framework that extends the previous amortized causal discovery work of Lowe et al..2022 where some extra assumptions of functional relationships are made to make it suitable for an edge-edge message passing framework on an GNN. Authors prove identifiability results for that specific modeling assumption.

**Concerns**: Concerns that were raised is that even in time series, non linear ICA results resolving the mixture of stationary time series were proven. So why is the current work different ? Authors responded by saying that their identifiability results do not require the support of the latent states to be known and relies on a 1968 result about finite mixtures of multi variate gaussians. Further, authors show empirically they outperform or match ACD which has lots of overlap in terms of architecture in many datasets.

Further, evaluations were not one sided with only comparisons with neural granger methods line of work alone but rather also includes R-PCMCI which is a constrained based Grange causal discovery method that does handle regime dependent time series.

**Overall**:  (I read the paper in some detail before coming to this conclusion) Authors very general identifiability results without knowing latent support and adaptation of GNN based VAE from prior work to solve for Granger causal relationships with conditionally stationary time series is noteworthy. Experimental evaluation (during rebuttal and in the paper) also look comprehensive. I encourage the authors to release their code base for future works to compare to theirs. I recommend accept